Article 

# Evolution of retinal degeneration and prediction of disease activity in relapsing and progressive multiple sclerosis

Julia Krämer [1] ✉, Carolin Balloff[2,3], Margit Weise[2], Valeria Koska[2], Yannik Uthmeier[2], Isabell Esderts[1], Mai Nguyen-Minh[1], Moritz Zimmerhof[1], Alex Hartmann [4], Michael Dietrich[2], Jens Ingwersen[2], John-Ih Lee[2], Joachim Havla[5], Tania Kümpfel[5], Martin Kerschensteiner [5,6,7], Vivien Häußler[8,9], Christoph Heesen[8,9], Jan-Patrick Stellmann[8,9,10,11], Hanna G. Zimmermann[12], Frederike C. Oertel[12], Marius Ringelstein [2,13], Alexander U. Brandt[12], Friedemann Paul[12], Orhan Aktas[2], Hans-Peter Hartung[2,14,15], Heinz Wiendl [1], Sven G. Meuth[2,16] & Philipp Albrecht [2,3,16] ✉

Retinal optical coherence tomography has been identified as biomarker for disease progression in relapsing-remitting multiple sclerosis (RRMS), while the dynamics of retinal atrophy in progressive MS are less clear. We investigated retinal layer thickness changes in RRMS, primary and secondary progressive MS (PPMS, SPMS), and their prognostic value for disease activity. Here, we analyzed 2651 OCT measurements of 195 RRMS, 87 SPMS, 125 PPMS patients, and 98 controls from five German MS centers after quality control. Peripapillary and macular retinal nerve fiber layer (pRNFL, mRNFL) thickness predicted future relapses in all MS and RRMS patients while mRNFL and ganglion cell-inner plexiform layer (GCIPL) thickness predicted future MRI activity in RRMS (mRNFL, GCIPL) and PPMS (GCIPL). mRNFL thickness predicted future disability progression in PPMS. However, thickness change rates were subject to considerable amounts of measurement variability. In conclusion, retinal degeneration, most pronounced of pRNFL and GCIPL, occurs in all subtypes. Using the current state of technology, longitudinal assessments of retinal thickness may not be suitable on a single patient level.

The chronic degeneration of inner retinal layers assessed by optical coherence tomography (OCT) has been established as a surrogate parameter not only for visual disability but also for global CNS degeneration in multiple sclerosis (MS)[1,2] correlating with clinical disability[1,3–6] and cognitive flexibility[7] and providing prognostic utility[4,5,8–21]. Volume changes of inner retinal layers were demonstrated to correlate with inflammatory disease activity and to indicate treatment response or non-response[15,22].

Some longitudinal investigations of retinal morphology in mixed relapsing and progressive MS cohorts have suggested an attenuated atrophy rate of the inner retinal layers with a longer disease duration[23,24].

In patients with progressive MS (PMS), retinal atrophy has so far only been examined in a few longitudinal studies comprising a comparatively small sample of patients and a short follow-up interval[23,25–27] and producing inconsistent results (retinal thinning[25,26] vs no thinning in PMS[23,27]). Recently, PMS was shown to be associated with faster retinal thinning independent of age, as compared to relapsing-remitting MS (RRMS) and healthy controls (HCs)[24].

While combined multimodal evoked potentials were demonstrated to correlate with and predict disability in mixed cohorts of RRMS and PMS patients[28,29], the extent and predictive value of visual evoked potentials (VEPs) in PMS were not investigated in longitudinal studies so far.

Against this background, we aimed to cross-sectionally and longitudinally investigate retinal layer thickness, visual acuity (VA), and VEP latency in relation to disease duration in patients with RRMS, primary and secondary progressive MS (PPMS, SPMS). Further objectives were to analyze the associations between retinal layer thickness, VA, and VEP latency, and their respective capability to predict clinical and radiological disease activity and disability progression in RRMS, SPMS, and PPMS.

Analyzing 2651 OCT measurements of 195 RRMS, 87 SPMS, 125 PPMS patients and 98 controls from five German MS centers, we could demonstrate that retinal degeneration occurred and predicted disease activity in all MS subtypes. However, longitudinal thickness change rates over reasonable intervals were subject to considerable amounts of measurement variability and not suitable to predict disease activity on a single patient level.

## Results

### Confounding ocular pathology and quality control

Rigorous control was applied to all assessments both on the eye level regarding confounding ocular pathology and on the measurement level regarding quality control (see Fig. 1 and Supplementary Table 1). Finally, 2651 measurements of 195 RRMS, 87 SPMS, 125 PPMS patients, and 98 HCs were included in the analysis (Fig. 1). 505 individuals ($N_{RRMS}$ = 195, $N_{SPMS}$ = 87, $N_{PPMS}$ = 125, $N_{HCs}$ = 98) had at least two OCT measurements, 263 individuals ($N_{RRMS}$ = 124, $N_{SPMS}$ = 41, $N_{PPMS}$ = 60, $N_{HCs}$ = 38) three OCT measurements, 132 individuals ($N_{RRMS}$ = 86, $N_{SPMS}$ = 13, $N_{PPMS}$ = 24, $N_{HCs}$ = 9) four OCT measurements, 64 individuals ($N_{RRMS}$ = 49, $N_{SPMS}$ = 6, $N_{PPMS}$ = 8, $N_{HCs}$ = 1) five OCT measurements, 29 individuals ($N_{RRMS}$ = 23, $N_{SPMS}$ = 3, $N_{PPMS}$ = 3, $N_{HCs}$ = 0) six OCT measurements, and 13 individuals ($N_{RRMS}$ = 11, $N_{SPMS}$ = 0, $N_{PPMS}$ = 2, $N_{HCs}$ = 0) seven OCT measurements.

### Study population

Baseline demographic and clinical data of patients and HCs are reported in Table 1. The following analyses were corrected for participants' age and/or sex if the model fit could be improved. This is indicated in all figures and/or tables presenting results.

### Retinal layer thickness, visual acuity, and VEP latency at baseline depending on the disease duration

Patients entered the study at different time points during the disease. Figure 2 displays the peripapillary retinal nerve fiber layer (pRNFL) (Fig. 2A, LMM), macular retinal nerve fiber layer (mRNFL) (Fig. 2B, LMM), ganglion cell-inner plexiform layer (GCIPL) (Fig. 2C, LMM), and inner nuclear layer (INL) thickness (Fig. 2D, LMM), the VA (Fig. 2E, LMM), and VEP latency (Fig. 2F, LMM) at baseline in relation to the disease duration for patients with RRMS, SPMS, and PPMS presenting coefficient estimates and 95% confidence intervals obtained from linear mixed-effects models (LMM). The pRNFL thickness was inversely associated with the disease duration in all subgroups (Fig. 2A, LMM). While the mRNFL and GCIPL thickness were inversely associated with the disease duration in all subgroups except PPMS (Fig. 2B, C, LMM), associations between the INL thickness and disease duration were only found in SPMS (Fig. 2D, LMM). VA at baseline was not associated with the disease duration in any subgroup (Fig. 2E, LMM) and the VEP latency was associated with the disease duration only in RRMS patients (Fig. 2F, LMM). In line with the results of the linear regression, spline fits demonstrated a decrease of pRNFL, mRNFL and even more GCIPL with increasing disease duration in the different MS subgroups (Supplementary Fig. 1, spline fit).

**709 subjects (1418 eyes) assessed for eligibility**
HC 119 subjects (238 eyes)
RRMS 261 subjects (522 eyes)
SPMS 139 subjects (278 eyes)
PPMS 190 subjects (380 eyes)

**542 eyes were excluded from the analysis due to different reasons:**
Insufficient OCT signal (N=101)
Drusen (N=51)
Myopia ≥ ±6 dpt (N=1)
Amblyopia (N=4)
RPE lesion (N=1)
Glaucoma (N=16)
Retinal hole (N=1)
ON ≤ 6 months before baseline or during FU (N=36)
Macular edema (N=12)
Retinitis pigmentosa (N=2)
Transition to SPMS during FU (N=6)
No fulfilled diagnosis criteria (N=4)
Missing clinical data (N=24)
FU ≤ 9 months (N=180)
Retinal pathology not further classified (N=13)
Epiretinal gliosis (N=22)
Macular dystrophy (N=2)
Incomplete OCT data (N=11)
OCT impossible due to fixation problems (N=2)
Traction phenomena (N=3)
Death (N=6)
Normal pressure hydrocephalus (N=2)
Maculopathy (N=3)
OCT under chemotherapy (N=2)
Retinal scar (N=3)
Branch retinal artery occlusion (N=1)
Macular hole and epiretinal membrane cystic lesion (N=1)
Bleedings and exudates (N=2)
CADASIL (N=2)
Wilson's disease (N=2)
Head tremor interfering with examination (N=2)
Foveal pathology (N=2)
Recurrent uveitis (N=4)
Iridocyclitis (N=2)
Macular infiltration (N=1)
Eccentric serous chorioretinopathy (N=1)
Papillary edema (N=2)
Amaurosis (N=1)
Peripapillary retinoschisis (N=1)

**Number of patients at baseline**
HC 98; RRMS 195; SPMS 87; PPMS 125
**Number of OCT measurements**
HC 453; RRMS 1175; SPMS 405; PPMS 618
**Number of OCT measurements at baseline**
HC 182; RRMS 326; SPMS 145; PPMS 223
**Number of OCT measurements at FU 1:**
HC 176; RRMS 336; SPMS 148; PPMS 224
**Number of OCT measurements at FU 2:**
HC 75; RRMS 216; SPMS 73; PPMS 107
**Number of OCT measurements at FU 3:**
HC 18; RRMS 147; SPMS 23; PPMS 41
**Number of OCT measurements at FU 4:**
HC 2; RRMS 87; SPMS 10; PPMS 15
**Number of OCT measurements at FU 5:**
RRMS 42; SPMS 6; PPMS 5
**Number of OCT measurements at FU 6:**
RRMS 21; PPMS 3
**Mean follow-up time (years; SD)**
HC 3.32 ± 2.59; RRMS 4.24 ± 2.12; SPMS 3.98 ± 2.63; PPMS 2.96 ± 1.61
**Number of VEP measurements**
HC 117; RRMS 595; SPMS 148; PPMS 191
**Number of visual testing**
HC 222; RRMS 411; SPMS 263; PPMS 381
**Number of MRI measurements**
HC: 60; RRMS 1634; SPMS 364; PPMS 478

**Fig. 1 | Flowchart of study design.** CADASIL Cerebral Autosomal Dominant Arteriopathy with Subcortical Infarcts and Leukoencephalopathy, FU follow-up, HC healthy controls, MRI magnetic resonance imaging, OCT optical coherence tomography, ON optic neuritis, PPMS primary progressive MS, RPE retinal pigment epithelium, RRMS relapsing-remitting multiple sclerosis, SD standard deviation, SPMS secondary progressive MS, VEP visual evoked potential.

**Table 1 | Baseline demographics and clinical characteristics**

| Characteristic | HCs | All MS | RRMS | PPMS | SPMS | P value |
|---|---|---|---|---|---|---|
| Subjects, n | 98 | 407 | 195 | 125 | 87[h] | |
| Female, n (%) | 64 (65.3) | 241 (59.2) | 126 (64.6) | 59 (47.2) | 56 (64.4) | 0.007[a] |
| Age at baseline, y, median (IQR) | 41 (30–51) | 45 (35–53) | 34 (28–42) | 50 (46–56) | 54 (46–59) | <0.001[b] |
| Disease duration at baseline, y, median (IQR) | | 5.2 (1.3–12.7) | 1.4 (0.6–4.6) | 6.9 (3.1–10.9) | 20.1 (13.4–25.4) | <0.001[c] |
| EDSS score at baseline, median (IQR) | | 3.0 (1.5–5) | 1.5 (1–2) | 4 (3–6) | 5.5 (4–6) | <0.001[d] |
| Eyes with history of ON[i] | | 112 | 59 | 11 | 42 | <0.001[e] |
| Eyes with clinical history of ON | | 80 | 44 | 0 | 36 | <0.001[e] |
| Eyes with history of ON identified by cut-off of GCIPL inter-eye difference | | 32 | 15 | 11 | 6 | <0.001[e] |
| On low-efficacy DMT at baseline, n (%) | | 82 (26.3) | 70 (45.2) | 6 (6.5) | 6 (9.4) | <0.001[f] |
| On high-efficacy DMT at baseline, n (%) | | 55 (17.6) | 27 (17.4) | 8 (8.6) | 20 (31.3) | <0.001[f] |
| Follow-up time, y, median (IQR) | 2.4 (1.2–3.7) | 3.2 (2.0–5.5) | 4.1 (2.4–5.9) | 2.6 (2.0–3.4) | 3.1 (1.7–6.3) | <0.001[g] |

*DMT* disease-modifying therapy, *EDSS* expanded disability status scale, *HCs* healthy controls, *IQR* inter-quartile range, *MS* multiple sclerosis, *n* number, *ON* optic neuritis, *PPMS* primary progressive multiple sclerosis, *RRMS* relapsing remitting multiple sclerosis, *SD* standard deviation, *SPMS* secondary progressive multiple sclerosis, *y* years.

[a]Chi-squared test (one-sided).

[b]Kruskal–Wallis test (one-sided); Pairwise comparisons revealed $p < 0.001$ for all comparisons except for PPMS vs SPMS ($p = 0.04$). Mann–Whitney-*U*-Test (two-sided) for HCs vs RRMS, for HCs vs SPMS, and for HCs vs PPMS revealed $p < 0.001$,

[c]Kruskal–Wallis test (one-sided); Pairwise comparisons: Pairwise comparisons revealed $p < 0.001$ for all comparisons.

[d]Kruskal–Wallis test (one-sided); Pairwise comparisons: SPMS vs RRMS ($p < 0.001$), SPMS vs PPMS ($p = 0.02$), RRMS vs PPMS ($p < 0.001$).

[e]Chi-squared test (one-sided); Pairwise comparisons revealed $p < 0.001$ for all comparisons.

[f]Chi-squared test (one-sided); Pairwise comparisons: Pairwise comparisons revealed $p < 0.001$ for all comparisons.

[g]Kruskal–Wallis-test (one-sided) and post-hoc Wilcoxon rank sum test (two-sided): HCs vs PPMS ($p = 0.49$), HCs vs RRMS ($p < 0.001$), HCs vs SPMS ($p = 0.27$), PPMS vs RRMS ($p < 0.001$), PPMS vs SPMS ($p = 0.27$), RRMS vs SPMS ($p = 0.43$). Values < 0.05 were deemed to be statistically significant. Corrections for type-I errors were performed using the Holm–Bonferroni method on the resulting *p* values.

[h]21 patients were active based on relapses, 7 patients were active based on MRI progression/activity, and 0 patients were active based on both relapses and MRI progression/activity in the year before baseline OCT.

[i]ON was identified by cut-off of GCIPL inter-eye difference and clinical documentation.

Adjusting the linear regressions between retinal layer thickness, VA, and VEP latency at baseline and disease duration additionally for optic neuritis (ON) besides participants' age and/or sex if the model fit could be improved, revealed the same findings for VA, VEP latency, mRNFL and GCIPL thickness (Supplementary Table 2, LMM). Associations between the INL thickness and disease duration were not found in SPMS and associations between pRNFL thickness and disease duration were not found in PPMS (Supplementary Table 2, LMM). Including an interaction term between disease duration and disease course, as well as participants' age sex, and/or and history of ON and checking its contrast-effects using an *F*-test revealed that the effect of baseline disease duration on retinal layers, VA, and latency did not differ between groups (pRNFL: $F(3,464) = 1.05$; $p = 0.37$; mRNFL: $F(3,487) = 0.99$; $p = 0.40$; GCIPL: $F(3,491) = 0.98$; $p = 0.40$; INL: $F(3,423) = 0.36$; $p = 0.78$; VEP latency: $F(3,183) = 0.26$; $p = 0.86$; VA: $F(3,203) = 1.55$; $p = 0.20$).

In addition, we have plotted the retinal layer thickness, visual acuity, and VEP latency at baseline depending on age (Supplementary Fig. 2, spline fit). Especially pRNFL and GCIPL thickness slowly decreased with increasing age in all MS subtypes (Supplementary Fig. 2, spline fit).

**Changes of retinal layer thickness, visual acuity, and VEP latency over time**

RRMS, SPMS, and PPMS patients all showed thickness loss of pRNFL, mRNFL, GCIPL, and INL throughout their disease course ($p < 0.05$) (Supplementary Table 4, LMM). A significant atrophy was not observed for INL in PPMS patients ($b = -0.02$; $p = 0.18$; Supplementary Table 4, LMM). Calculating the median and IQR revealed the most pronounced changes in pRNFL and GCIPL (median (IQR) $pRNFL_{RRMS}$ $-0.49\,\mu m$ $(-1.44$–$0.00)$; $pRNFL_{SPMS}$ $-0.18\,\mu m$ $(-1.03$–$0.64)$; $pRNFL_{PPMS}$ $-0.39\,\mu m$ $(-1.33$–$0.00)$; $mRNFL_{RRMS}$ $0.00\,\mu m$ $(-0.67$–$0.38)$; $mRNFL_{SPMS}$ $0.00\,\mu m$ $(-0.35$–$0.35)$; $mRNFL_{PPMS}$ $-0.07\,\mu m$ $(-0.68$–$0.34)$; $GCIPL_{RRMS}$ $-0.18\,\mu m$ $(-0.68$–$0.23)$; $GCIPL_{SPMS}$ $-0.20\,\mu m$ $(-0.55$–$0.00)$; $GCIPL_{PPMS}$ $-0.33\,\mu m$ $(-0.70$–$0.00)$; $INL_{RRMS}$ $0.00\,\mu m$ $(-0.35$–$0.32)$; $INL_{SPMS}$ $0.00\,\mu m$ $(-0.35$–$0.27)$; $INL_{PPMS}$ $0.00\,\mu m$ $((-0.35$–$0.31))$) (Supplementary Table 3). However, the overall annualized thickness change rate over time was very low, as indicated by boxplots bordered in red in Fig. 3A–D. These boxplots bordered in red represent the average of all annualized thickness change rates across time. Every single annualized thickness change rate was defined as the difference between two consecutive OCT measurements divided by the time between measurements in years (Fig. 3A–D).

Because patients participated in different observational studies at the five MS centers, follow-up assessments were performed at varying time points and with different frequencies. To account for the varying influence of disease duration at the different stages of the disease, the RRMS, SPMS, and PPMS samples were divided into six to seven subgroups according to the disease duration for further longitudinal analysis (Fig. 3, colored boxplots depicting annualized thickness change rates as difference between two consecutive OCT measurements divided by the time between measurements in years). In HCs, we considered the time since baseline OCT.

Significant loss (LMM analysis) of pRNFL and GCIPL thickness occurred throughout the disease course in RRMS and PPMS while in SPMS pRNFL thickness decreased early (<12.6 years) and GCIPL thickness late (>25.6 years) (Fig. 3A, C and Supplementary Table 5, LMM). The thickness change of the other retinal layers was quite heterogeneous for the different disease duration intervals. mRNFL atrophy was observed in the early phases of disease in RRMS (<3.5 years) and in the later stages in RRMS and PPMS (>10.6 years) and SPMS (>25.6 years) (Fig. 3B, Supplementary Table 5, LMM), INL atrophy early in the disease course in PPMS (<7.5 years) and late in SPMS (>30.6 years) after previous thickening in SPMS (16.6–20.5 years) (Fig. 3D, Supplementary Table 5, LMM). We found no effect of the disease duration on the VA in any group (RRMS: $p = 0.17$, $b = 0.003$; PPMS: $p = 0.82$, $b = -0.0006$; SPMS $p = 0.41$, $b = 0.002$), while the disease duration had significant influence on the VEP latency in the RRMS and

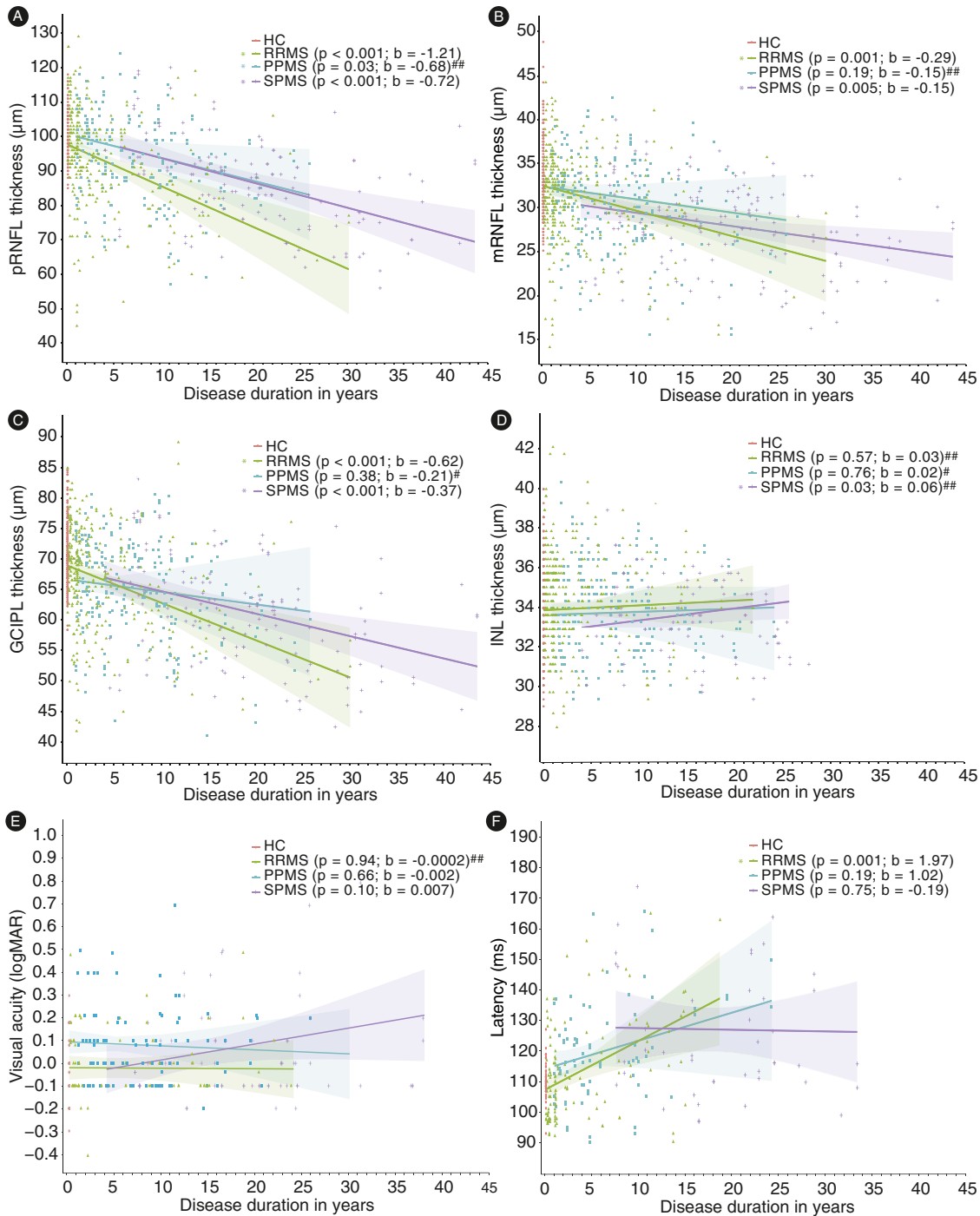

**Fig. 2 | Linear regressions between retinal layer thickness, visual acuity, and VEP latency at baseline and disease duration.** Analyses were carried out with linear mixed-effects models (LMM). A separate model was calculated for each group. The results of the different assessments are plotted against the disease duration at baseline, each dot representing a single eye. Coefficient estimates and linear regression lines with 95% confidence intervals obtained from LMM are provided for the different subgroups. The measure of center for the error bands is the predicted value for the outcome (**A**) The pRNFL thickness was inversely associated with the disease duration in all subgroups. **B**, **C** The mRNFL and GCIPL thickness were inversely associated with the disease duration in RRMS and SPMS. **D** The INL thickness was associated with the disease duration in SPMS. **E** The visual acuity at baseline was not associated with the disease duration in any subgroup. **F** The VEP latency was associated with the disease duration in RRMS. The disease duration corresponded the time since baseline OCT in healthy controls and had a value of 0. # control variable age was added to the model to improve the model fit ## control variable sex was added to the model to improve the model fit.

PPMS group (RRMS: $p < 0.001$, $b = 0.80$; PPMS: $p = 0.03$, $b = 0.53$; SPMS $p = 0.08$, $b = 0.52$). The variables age and/or sex did not improve the model fit.

In line with the results of the mixed linear regression models, spline fits demonstrated decrease of pRNFL, mRNFL and even more GCIPL in the different MS subgroups (Fig. 3A–C, spline fit). For reasons

of completeness, we also plotted longitudinal changes of retinal layer thickness, visual acuity, and VEP latency over time depending on age for the different subgroups (Supplementary Fig. 3, spline fit). Spline fits demonstrated decreasing pRNFL and GCIPL thickness with increasing age in HCs and the different MS subgroups (Supplementary Fig. 3, spline fit).

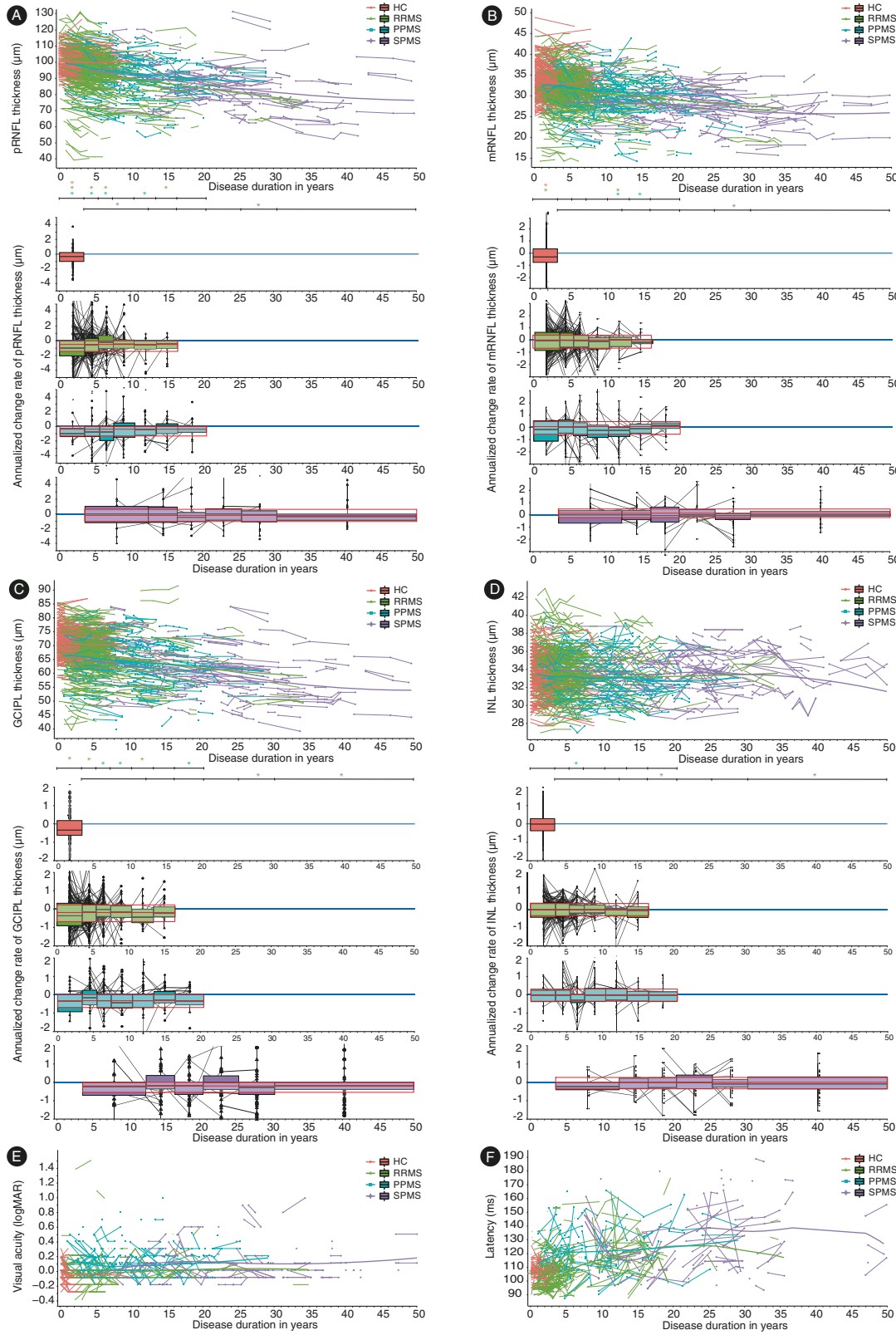

Empirical Bayes estimates of best linear unbiased predictions (BLUPs) of annualized thickness change rates demonstrated significant atrophy of all retinal layers (pRNFL, mRNFL, GCIPL, and INL) for all groups (Supplementary Fig. 4, BLUPs) and analysis of variance (ANOVA) significant differences across disease courses for pRNFL ($F_{(2,367)} = 3.65$; $p = 0.03$) and mRNFL ($F_{(2,404)} = 4.43$; $p = 0.01$) (Supplementary Table 7). A Tukey post-hoc test revealed higher atrophy rates of pRNFL for RRMS (diff = 0.11, $p = 0.02$) compared to SPMS and of mRNFL for PPMS (diff = 0.06, $p = 0.01$) compared to SPMS.

Wilcoxon rank-sum tests of eye wise ordinary least squares (OLS) coefficients for disease duration with retinal layer thickness as outcome variable demonstrated significant atrophy of all retinal layers of all groups, except for SPMS with regard to pRNFL, mRNFL, and INL and PPMS with regard to INL (Supplementary Fig. 5). ANOVA of eye wise

**Fig. 3 | Changes of retinal layer thickness (A–D), visual acuity (E), and VEP latency (F) over time.** On the top, the raw data of retinal layer thickness (**A**: pRNFL, **B**: mRNFL, **C**: GCIPL; **D**: INL) are plotted over the disease duration, each dot representing a single eye and follow-up assessments being connected with lines. For HCs the disease duration was plotted as time since baseline OCT. B-splines were fitted in a mixed model with random intercept for eyes and controlling for gender. The number of knots and polynominal degrees (both between 1 and 5) were chosen such that 10-fold cross validation mean squared error (MSE) was minimized. In order to achieve an equal distribution of data within the different intervals the boundaries had to be set differently for SPMS than for RRMS and PPMS. The following intervals of disease duration were used: 0–3.5 years for HCs, 0–3.5 years, 3.6–5.5 years, 5.6–7.5 years, 7.6–10.5 years, 10.6–13.5 years, 13.6–16.5 years for PPMS and RRMS patients, additionally 16.6–20.5 years for PPMS, and 3.5–12.5 years, 12.6–16.5 years, 16.6–20.5 years, 20.6–25.5 years, 25.6–30.5 years, and over 30.6 years for SPMS patients. The sample sizes for each interval of disease duration by disease subtype is reported in Supplementary Table 6 (see Supplementary Information). The time interaction indicating thickness changes over time was analyzed separately for the time intervals displayed on the horizontal bar below the scatter plots and significant changes ($p < 0.05$, LMM) are indicated with asterisks of the color matching the group. The $p$ and $b$ values (including 95% confidence intervals) are displayed in Supplementary Table 5, LMM (see Supplementary Information). The colored boxplots below indicate the annualized thickness change rate of the different layers for the different intervals. For each time of assessment, the annualized thickness change rate since the last assessment was calculated based on the following formula: annualized thickness change rate= (retinal layer thickness$_{current\ assessment}$ - retinal layer thickness$_{previous\ assessment}$) / time since last assessment in years). The boxplots, which are outlined in red, show the average of all annualized thickness change rates over time of the different layers (**A–D**). The bounds of box plots are the IQR and the measure of center is the median. $P$ and $b$ values (including 95% confidence intervals) obtained from the LMM analysis can be found in Supplementary Table 4 (see Supplementary Information). The raw data of visual acuity (**E**) and VEP latency (**F**) are plotted over the disease duration for the different subgroups, each dot representing an eye and follow up measurements being connected by lines. Because we wanted to avoid power issues, no intervals for different periods of disease duration were created.

OLS atrophy rate estimations with random intercept per subject showed significant differences of atrophy rates across disease courses for pRNFL ($F_{(2186)} = 4.14$; $p = 0.02$) and mRNFL ($F_{(2215)} = 4.34$; $p = 0.01$) (Supplementary Table 8). A Tukey-test revealed lower atrophy rates of pRNFL for SPMS compared to RRMS (diff = −0.39, $p = 0.02$) and PPMS (diff = −0.43, $p = 0.03$) and of mRNFL for SPMS compared to PPMS (diff = −0.26, $p = 0.01$).

### Associations between retinal layer thickness and visual acuity, and VEP latency
VA was significantly associated with the pRNFL thickness in RRMS and SPMS (Fig. 4A, LMM) and with the mRNFL and GCIPL thickness in all MS groups (Fig. 4B, C, LMM). The INL thickness was not associated with the VA in any group (Fig. 4D, LMM).

VEP latency was significantly associated with the pRNFL and GCIPL thickness in all MS groups (Fig. 4E, F, LMM), with the mRNFL thickness in HCs, RRMS and SPMS patients (Fig. 4G, LMM), and with the INL thickness only in SPMS patients (Fig. 4H, LMM).

### Prediction of disease activity by retinal layer thickness
To analyze the potential of retinal layer thicknesses to predict disability progression (Expanded Disability Status Scale (EDSS)[30] worsening), MRI progression/activity (new or enlarging T2-weighted/gadolinium-enhancing lesions), and relapses, we used logistic mixed-effects regression (LMER) (Tables 2–4), Kaplan–Meier analyses (Fig. 5) and Cox regression models (Supplementary Table 9) while adjusting for age, sex, EDSS, and/or DMT. Moreover, logistic regressions were adjusted for the time to assessment of EDSS, MRI, and relapses (see methods).

Of all patients included in the logistic mixed-effects regression analysis, 47% RRMS and 28% SPMS patients experienced a relapse during their mean time (SD) from OCT to relapse assessment of 1.6 (±1.2) years and 57% RRMS, 22% SPMS, and 16% PPMS patients had MRI progression/activity during their mean time (SD) from OCT to MRI assessment of 1.5 (±1.3) years. 27% RRMS, 30% SPMS, and 33% PPMS had EDSS worsening until their last follow-up visit.

When adjusting for all covariates in the LMER, lower pRNFL and mRNFL thickness were associated with increased probability for relapses in all MS and RRMS patients without ON (pRNFL and mRNFL) and all RRMS patients (pRNFL) (Table 2, LMER). mRNFL and GCIPL thickness predicted future MRI progression/activity in RRMS without ON (mRNFL and GCIPL thickness) and PPMS patients (GCIPL thickness) (Table 4, LMER). When adjusting for age, sex, and DMT in the LMER, mRNFL thickness predicted future disability progression in PPMS (Table 3, LMER).

The significance levels and odds ratios are provided in Tables 2–4. As an example, 1 μm of pRNFL thickness loss in RRMS patients without ON increases the likelihood of relapse by 34% (Risk factor, Table 2, LMER).

Kaplan–Meier analyses and Cox regression models confirmed the result that PPMS patients with lower GCIPL thickness (Table 4, LMER) or mRNFL, GCIPL, and INL thickness of the lowest tertile (Cox model: mRNFL ≤ 29 μm, 17 of 54, 31%; GCIPL ≤ 62 μm, 18 of 54, 33%; INL ≤ 33 μm, 24 of 58, 41%) had significantly increased rates of subsequent MRI progression/activity (Cox model: lowest versus the two upper tertiles: mRNFL: HR, 2.81 [95% CI: 1.13–7.01]; $p = 0.03$; GCIPL: HR, 2.61 [95% CI: 1.05–6.47]; $p = 0.04$; INL: HR, 2.53 [95% CI: 1.03–6.21]; $p = 0.04$); Fig. 5 and Supplementary Table 9).

### Prediction of disease activity by VEP latency
To analyze the potential of VEP latency to predict disability progression (EDSS worsening), MRI progression/activity (new or enlarging T2-weighted/gadolinium-enhancing lesions), and relapses, we used logistic mixed-effects regression while adjusting for disease duration or age, ON, sex, EDSS, DMT, and time to assessment of EDSS, MRI, and relapses (see methods). The analyses did not reveal conclusive results as very wide confidence intervals resulted from a combination of the small sample size per stratum and the high variability in covariates.

### Prediction of disease activity by OCT thickness change rates
To analyze the potential of longitudinally assessed thickness change rates of pRNFL, mRNFL, GCIPL, and INL to predict disability progression (EDSS worsening), MRI progression/activity (new or enlarging T2-weighted/gadolinium-enhancing lesions), and relapses, we used logistic mixed-effects regression while adjusting for disease duration or age, ON, sex, EDSS, DMT, and time to assessment of EDSS, MRI, and relapses (see methods). The analyses did not reveal conclusive results as very wide confidence intervals resulted from a combination of the small sample size per stratum and the high variability in covariates.

## Discussion
We present a large longitudinal multicenter study analyzing changes of retinal layer thickness and visual function and their predictive power for subsequent disease activity and disability progression in RRMS, SPMS, and PPMS patients.

Our study shows that pRNFL and GCIPL decreased throughout the disease course in both relapsing and progressive MS while the other retinal layers presented a more heterogeneous atrophy pattern (Fig. 3A–D). Our findings corroborate and expand upon previous findings[23,26,31] demonstrating that the pRNFL thickness loss was most pronounced at the beginning of disease and diminished with longer disease duration not only in relapsing but also in primary progressive MS (Fig. 3A). GCIPL atrophy rates were

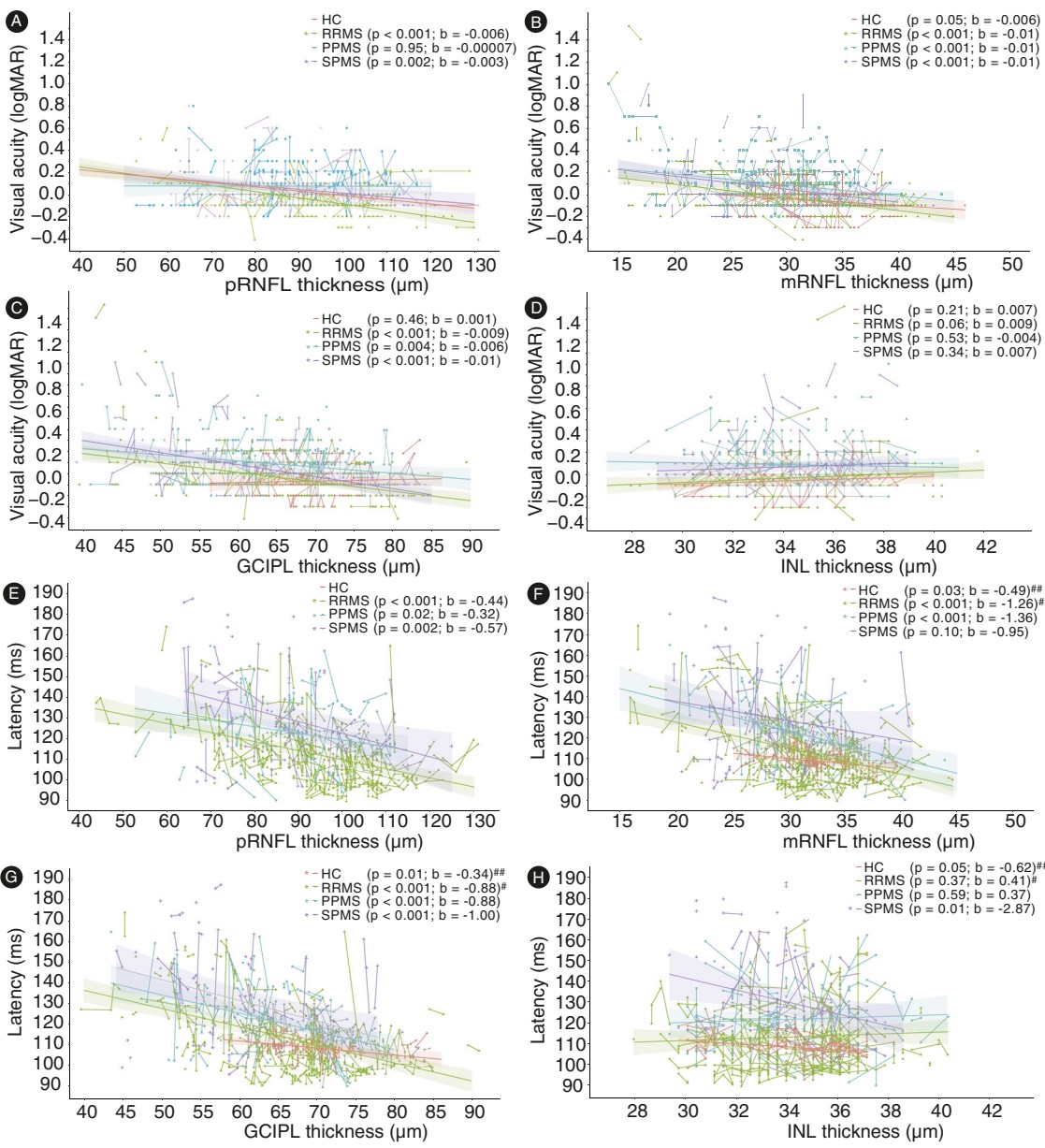

**Fig. 4 | Regressions between retinal layer thickness, visual acuity, and VEP latency over time.** All analyses were carried out using LMM. A separate model was calculated for each group. The visual acuity (**A**–**D**) and VEP latency (**E**–**H**) data are plotted over the thickness of the different retinal layers, each dot representing one eye and follow-up assessments being connected with lines. Linear regression lines with 95% confidence interval are provided for the different subgroups. The measure of center for the error bands is the predicted value for the outcome (**A**) The visual acuity was inversely associated with the pRNFL thickness in RRMS and SPMS, (**B**) the mRNFL, and (**C**) GCIPL thickness in RRMS, SPMS, and PPMS. **E** The VEP latency was inversely associated with the pRNFL thickness in RRMS, SPMS, and PPMS, (**F**) the mRNFL in HCs, RRMS, and PPMS, (**G**) the GCIPL thickness in RRMS, SPMS, PPMS, and HCs (**H**), and the INL thickness in SPMS. # control variable age was added to the model to improve the model fit; ## control variable sex was added to the model to improve the model fit.

highest at the beginning and in later phases of disease in relapsing and progressive MS (Fig. 3C). Interestingly, the INL thickness only decreased in the earlier phases of PPMS (disease duration of 5.6–7.5 years) and late phases of SPMS (disease duration over 30.6 years) but not in RRMS (Fig. 3D). While INL thickening was previously reported to be associated with recent and subsequent clinical and radiological disease activity in RRMS[15,32–34], INL atrophy was found in people with longstanding and/or progressive MS disease[1,3,35]. Our findings are in line with the assumption that INL swelling during the more inflammatory phases in RRMS is ultimately followed by INL atrophy and neuronal loss in the later phases of SPMS while PPMS patients already present with a prominent degenerative pathology at the time of diagnosis.

The overall annualized thickness change rate over time was very low and most pronounced in pRNFL and GCIPL (Fig. 3A–D and Supplementary Table 3). These thickness change rates are comparable with previous studies[4,23,24]. Our longitudinal assessments, which were performed in the clinical routine and using the current standard of technology, were subject to a considerable amount of measurement variability often exceeding the change rates observed in the cohort. This has direct implications for the clinical routine as it limits the usefulness of longitudinal assessments to investigate change rates for prognostic purposes on a single subject level. Further developments of technology, including OCT hardware, optics, quality control and post-acquisition analysis are already underway and may help to decrease measurement variability in the future.

**Table 2 | Prediction of future disease activity (relapses) by retinal layer thickness**

| | Relapse | | | |
|---|---|---|---|---|
| | **Odds ratio** | **p-value** | **Prob. of event** | **Risk factor** |
| pRNFL | | | | |
| All patients | 0.639 (0.361–1.129)[a] | 0.123[b,c,d,e] | 19.51% (262/1343) | 1.092 |
| All patients w/o ON | **0.299 (0.144–0.624)[a]** | **0.001[b,c,d,e]** | **23.5% (153/651)** | **1.334** |
| RRMS | **0.552 (0.323–0.94)** | **0.029[b,c,d,e]** | **24.41% (198/811)** | **1.127** |
| RRMS w/o ON | **0.297 (0.14–0.63)** | **0.002[b,c,d,e]** | **23.48% (127/541)** | **1.34** |
| SPMS | 1.15 (0.353–3.745) | 0.817[c,e] | 28.32% (64/226) | 0.971 |
| SPMS w/o ON | 2.601 (0.084–80.84) | 0.586[c,d,e] | 23.64% (26/110) | 0.772 |
| PPMS | – | – | – | – |
| mRNFL | | | | |
| All patients | 0.539 (0.276–1.053)[a] | 0.071[b,c,d,e] | 19.13% (234/1223) | 1.044 |
| All patients w/o ON | **0.353 (0.154–0.811)[a]** | **0.014[b,c,d,e]** | **22.87% (142/621)** | **1.088** |
| RRMS | 0.616 (0.352–1.077) | 0.089[b,c,d,e] | 23.42% (178/760) | 1.034 |
| RRMS w/o ON | **0.305 (0.133–0.702)** | **0.005[b,c,d,e]** | **22.65% (118/521)** | **1.105** |
| SPMS | 0.998 (0.233–4.281) | 0.998[c,d,e] | 27.86% (56/201) | 1 |
| SPMS w/o ON | 0.409 (0.001–154.283) | 0.768[b,c,d,e] | 24% (24/100) | 1.07 |
| PPMS | – | – | – | – |
| GCIPL | | | | |
| All patients | 0.68 (0.362–1.279)[a] | 0.232[b,c,d,e] | 19.51% (262/1343) | 1.046 |
| All patients w/o ON | 0.602 (0.27–1.341)[a] | 0.214[b,c,d,e] | 23.5% (153/651) | 1.072 |
| RRMS | 0.648 (0.366–1.149) | 0.138[b,c,d,e] | 24.41% (198/811) | 1.053 |
| RRMS w/o ON | 0.432 (0.184–1.018) | 0.055[b,c,d,e] | 23.48% (127/541) | 1.127 |
| SPMS | 1.019 (0.287–3.614) | 0.976[c,e] | 28.32% (64/226) | 0.998 |
| SPMS w/o ON | 6.911 (0.073–651.191) | 0.405[b,c,d,e] | 23.64% (26/110) | 0.768 |
| PPMS | – | – | – | – |
| INL | | | | |
| All patients | 0.986 (0.501–1.942)[a] | 0.968[b,c,d,e] | 19.51% (262/1343) | 1.006 |
| All patients w/o ON | 1.321 (0.655–2.666)[a] | 0.437[b,c,d,e] | 23.5% (153/651) | 0.881 |
| RRMS | 0.903 (0.487–1.675) | 0.746[b,c,d,e] | 24.41% (198/811) | 1.044 |
| RRMS w/o ON | 1.307 (0.641–2.667) | 0.462[b,c,d,e] | 23.48% (127/541) | 0.885 |
| SPMS | 1.043 (0.285–3.822) | 0.949[c,e] | 28.32% (64/226) | 0.98 |
| SPMS w/o ON | 2.027 (0.04–103.859) | 0.725[c,d,e] | 23.64% (26/110) | 0.717 |
| PPMS | – | – | – | – |

Bold values indicate statistical significance p < 0.05.
LMER was used to calculate the probability of events for all measurement time points on the eye level displayed for pRNFL, mRNFL, GCIPL, and INL thickness for all patients (RRMS, SPMS, and PPMS), for all RRMS and those without previous ON, all SPMS and those without previous ON, and all PPMS. All statistical tests were two-sided. Without previous ON means without ON based on medical history and without ON based on inter-eye GCIPL thickness difference[43]. Mean time (SD) to relapse assessment from OCT to post-OCT relapse: 1.6 y (±1.2 y). w/o = without. Risk factor = factor by which the risk of relapses increases for each μm thickness loss. Risk factor = $\frac{1}{OR^{\frac{1}{7}}}$ The following covariates were included in the feature selection process for the LMER: Age; TTA=time to assessment; EDSS; DMT; Sex.
[a]excluding patients with PPMS.
[b]Age.
[c]TTA = time to assessment.
[d]EDSS.
[e]DMT.

On the group level, we attempt to counter the measurement variability by using robust estimation techniques. Both eye-wise OLS and empirical Bayes estimates of BLUPs showed significant atrophy of all retinal layers for all groups (Supplementary Fig. 4). Exceptions to this rule were SPMS patients, who did not show significant longitudinal effects of disease duration on pRNFL, mRNFL and INL and PPMS patients on INL (Supplementary Fig. 5).

Besides retinal atrophy in all MS subtypes at different phases of disease, the group of HCs also showed pRNFL, mRNFL, and GCIPL thickness loss over a mean follow-up of 3.5 years. The amount of retinal atrophy observed in our HCs (boxplots in Fig. 3A–D) is in line with previous studies[23,24,26]. A recent study demonstrated that with increasing age, the rate of pRNFL atrophy in MS approaches rates

similar to those expected with normal aging[24], which is in line with our findings. Histopathological studies on human retina have demonstrated a decrease in the density of photoreceptors, ganglion cells, and retinal pigment epithelial cells with increasing age[36].

In contrast to the OCT measurements, VA did not correlate with disease duration in any MS subtype while VEP latency was only associated with disease duration in RRMS (Fig. 2E, F). This may suggest that the sensitivity of structural readouts assessed by OCT for detecting influence of disease duration on visual pathway pathology is superior to the sensitivity of functional readouts assessed by VEP and VA. We have to acknowledge that the sample size for the VEP assessments in our study was lower than for OCT. However, VEPs as performed in the clinical routine at the centers were subject to an even higher degree of

**Table 3 | Prediction of future disease activity (EDSS progression) by retinal layer thickness**

| | EDSS Progression | | | |
|---|---|---|---|---|
| | **Odds ratio** | ***p*-value** | **Prob. of event** | **Risk factor** |
| **pRNFL** | | | | |
| All patients | 0.978 (0.373–2.566) | 0.963[a] | 31.42% (192/611) | 1.004 |
| All patients w/o ON | 1.074 (0.3–3.846) | 0.913[a] | 28.36% (116/409) | 0.985 |
| RRMS | 1.035 (0.245–4.37) | 0.963[a] | 27.33% (82/300) | 0.993 |
| RRMS w/o ON | 1.48 (0.166–13.183) | 0.725[a] | 22.28% (43/193) | 0.911 |
| SPMS | 1.079 (0.146–7.955) | 0.941[a] | 35.71% (45/126) | 0.985 |
| SPMS w/o ON | 0.905 (0.047–17.313) | 0.947[a] | 34.43% (21/61) | 1.026 |
| PPMS | 0.969 (0.163–5.758) | 0.973[a,b] | 35.14% (65/185) | 1.006 |
| **mRNFL** | | | | |
| All patients | 0.915 (0.322–2.601) | 0.867[a] | 30.51% (169/554) | 1.006 |
| All patients w/o ON | 0.966 (0.249–3.752) | 0.961[a] | 28.16% (107/380) | 1.003 |
| RRMS | 1.039 (0.227–4.757) | 0.961[a] | 26.55% (73/275) | 0.997 |
| RRMS w/o ON | 1.202 (0.145–9.995) | 0.865[a] | 22.7% (42/185) | 0.985 |
| SPMS | 0.963 (0.098–9.498) | 0.974[a] | 34.48% (40/116) | 1.003 |
| SPMS w/o ON | 0.975 (0.035–27.431) | 0.988[a] | 34.48% (20/58) | 1.002 |
| PPMS | **0 (0–0.028)** | **0.002[a]** | **34.36% (56/163)** | **2.195** |
| **GCIPL** | | | | |
| All patients | 0.927 (0.336–2.552) | 0.883[a] | 31.42% (192/611) | 1.009 |
| All patients w/o ON | 0.929 (0.245–3.519) | 0.914[a] | 28.36% (116/409) | 1.010 |
| RRMS | 0.964 (0.219–4.238) | 0.961[a] | 27.33% (82/300) | 1.004 |
| RRMS w/o ON | 1.057 (0.12–9.274) | 0.96[a] | 22.28% (43/193) | 0.992 |
| SPMS | 1.104 (0.14–8.732) | 0.925[a] | 35.71% (45/126) | 0.989 |
| SPMS w/o ON | 0.798 (0.034–18.847) | 0.889[a] | 34.43% (21/61) | 1.028 |
| PPMS | 0.837 (0.14–4.991) | 0.845[a] | 35.14% (65/185) | 1.022 |
| **INL** | | | | |
| All patients | 0.738 (0.244–2.227) | 0.589[a] | 31.42% (192/611) | 1.142 |
| All patients w/o ON | 0.673 (0.148–3.063) | 0.608[a] | 28.36% (116/409) | 1.193 |
| RRMS | 0.697 (0.129–3.77) | 0.675[a] | 27.33% (82/300) | 1.164 |
| RRMS w/o ON | 0.6 (0.046–7.759) | 0.696[a] | 22.28% (43/193) | 1.259 |
| SPMS | 0.666 (0.061–7.259) | 0.738[a] | 35.71% (45/126) | 1.204 |
| SPMS w/o ON | 0.625 (0.015–26.02) | 0.805[a] | 34.43% (21/61) | 1.231 |
| PPMS | 0.823 (0.126–5.369) | 0.839[a] | 35.14% (65/185) | 1.092 |

Bold values indicate statistical significance $p < 0.05$.
LMER was used to calculate the probability of events for all measurement time points on the eye level displayed for pRNFL, mRNFL, GCIPL, and INL thickness for all patients (RRMS, SPMS, and PPMS), for all RRMS and those without previous ON, all SPMS and those without previous ON, and all PPMS. All statistical tests were two-sided. Without previous ON means without ON based on medical history and without ON based on inter-eye GCIPL thickness difference[43]. w/o = without. Risk factor = factor by which the risk of EDSS progression increases for each μm thickness loss. Risk factor $= \frac{1}{OR^{\frac{1}{5}}}$

The following covariates were included in the feature selection process for the LMER: Age; TTA=time to assessment; EDSS; DMT; Sex.
[a]Age.
[b]Sex.

variability than OCT measures suggesting that even with higher sample sizes the predictive value would not increase. We conclude that more advanced measures of standardization and quality control need to be implemented for VEP, including but not limited to cut-offs for minimum amplitudes and signal-to-noise ratios. The missing correlation between full contrast VA and disease duration in our study (Fig. 2) supports the notion that the full contrast VA as measured in the routine clinical neurology is not sensitive enough to detect subtle visual pathway involvement in MS[37]. Low contrast VA has been demonstrated to be superior to full contrast VA[38]. However, unfortunately low contrast VA was not available for the majority of participants.

In order to analyze the effect of time (disease duration) on retinal layer thickness, we present multiple approaches, each providing a unique perspective on estimation of retinal thickness change rates. With the baseline LMMs (Fig. 2), we present an exclusively cross-sectional approach to estimate thickness change rates and

complement it using non-linear spline regressions to capture potential non-linear thickness change rates (Supplementary Fig. 1). Furthermore, we present two approaches that measure only longitudinal effects, with the BLUPs being based on an overall LMM (Supplementary Fig. 4, Supplementary Table 7) and the eye-wise OLS on separate models for each eye (Supplementary Fig. 5, Supplementary Table 8). Thus, BLUPs of thickness change rates consider a larger sample by also respecting overall thickness change rates, while the OLS approach focuses exclusively on estimating overall thickness change rates for each eye separately. Therefore, BLUPs provide a lower variance estimation of thickness change, which also causes smaller differences in estimates across patients. These differences in thickness change are better captured by eye-wise OLS, which in turn suffers from higher variance in estimates.

In addition to providing both complementary views on the estimation of thickness change rates (longitudinal and cross-sectional), we

**Table 4 | Prediction of future disease activity (MRI progression/activity) by retinal layer thickness**

| | MRI progression/activity | | | |
| | Odds ratio | *p*-value | Prob. of event | Risk factor |
|---|---|---|---|---|
| **pRNFL** | | | | |
| All patients | 0.947 (0.755–1.188) | 0.639[a,b,c] | 25.33% (460/1816) | 1.011 |
| All patients w/o ON | 1.1 (0.81–1.495) | 0.541[a,b,c] | 24.46% (297/1214) | 0.977 |
| RRMS | 0.943 (0.733–1.214) | 0.648[a,b,c] | 30.44% (386/1268) | 1.013 |
| RRMS w/o ON | 0.968 (0.686–1.366) | 0.853[a,b,c] | 30.09% (254/844) | 1.008 |
| SPMS | 1.28 (0.693–2.365) | 0.431[a] | 14.23% (37/260) | 0.942 |
| SPMS w/o ON | 1.565 (0.408–6.004) | 0.514[a] | 11.11% (14/126) | 0.862 |
| PPMS | 0.766 (0.417–1.404) | 0.388[a] | 12.85% (37/288) | 1.063 |
| **mRNFL** | | | | |
| All patients | 0.99 (0.779–1.258) | 0.934[a,b,c] | 24.36% (409/1679) | 1.001 |
| All patients w/o ON | 1.249 (0.898–1.738) | 0.187[a,b,c] | 23.53% (269/1143) | 0.981 |
| RRMS | 1.033 (0.791–1.348) | 0.814[a,b,c] | 29.41% (350/1190) | 0.998 |
| **RRMS w/o ON** | **1.465 (1.026–2.091)** | **0.036[a,b]** | **29.01% (235/810)** | **0.968** |
| SPMS | 1.436 (0.741–2.781) | 0.284[a] | 13.3% (31/233) | 0.97 |
| SPMS w/o ON | 1.492 (0.358–6.215) | 0.583[a] | 12.07% (14/116) | 0.959 |
| PPMS | 0.528 (0.263–1.063) | 0.074[a] | 10.94% (28/256) | 1.059 |
| **GCIPL** | | | | |
| All patients | 1.013 (0.789–1.301) | 0.92[a,b,c] | 25.33% (460/1816) | 0.998 |
| All patients w/o ON | 1.227 (0.875–1.719) | 0.236[a,b,c] | 24.46% (297/1214) | 0.972 |
| RRMS | 1.121 (0.85–1.478) | 0.418[a,b,c] | 30.44% (386/1268) | 0.986 |
| **RRMS w/o ON** | **1.481 (1.02–2.152)** | **0.039[b,c]** | **30.09% (254/844)** | **0.947** |
| SPMS | 1.242 (0.641–2.408) | 0.521[a] | 14.23% (37/260) | 0.975 |
| SPMS w/o ON | 1.496 (0.263–8.508) | 0.65[a] | 11.11% (14/126) | 0.942 |
| **PPMS** | **0.475 (0.25–0.905)** | **0.024[a]** | **12.85% (37/288)** | **1.114** |
| **INL** | | | | |
| All patients | 1.067 (0.819–1.39) | 0.63[a,b,c] | 25.33% (460/1816) | 0.973 |
| All patients w/o ON | 1.055 (0.773–1.44) | 0.737[a,b,c] | 24.46% (297/1214) | 0.977 |
| RRMS | 1.043 (0.769–1.415) | 0.786[a,b,c] | 30.44% (386/1268) | 0.983 |
| RRMS w/o ON | 1.1 (0.785–1.542) | 0.58[a,b,c] | 30.09% (254/844) | 0.958 |
| SPMS | 1.825 (0.828–4.025) | 0.136[a] | 14.23% (37/260) | 0.765 |
| SPMS w/o ON | 4.439 (0.935–21.079) | 0.061[a] | 11.11% (14/126) | 0.533 |
| PPMS | 0.704 (0.39–1.268) | 0.242[a] | 12.85% (37/288) | 1.18 |

Bold values indicate statistical significance *p* < 0.05.
LMER was used to calculate the probability of events for all measurement time points on the eye level displayed for pRNFL, mRNFL, GCIPL, and INL thickness for all patients (RRMS, SPMS, and PPMS), for all RRMS and those without previous ON, all SPMS and those without previous ON, and all PPMS. All statistical tests were two-sided. Without previous ON means without ON based on medical history and without ON based on inter-eye GCIPL thickness difference[43]. Mean time (SD) to MRI assessment from OCT to post-OCT MRI: 1.5 y (±1.3 y); w/o = without. Risk factor = factor by which the risk of MRI progression/activity increases for each µm thickness loss. Risk factor $= \frac{1}{OR^{\frac{1}{2}}}$. The following covariates were included in the feature selection process for the LMER: Age; TTA=time to assessment; EDSS; DMT; Sex.
[a]Age.
[b]TTA = time to assessment.
[c]EDSS.

also present models that include both types of effects. Namely, we present a regular LMM (Supplementary Table 4) estimating linear thickness change rates across the entire sample as well as a LMM with the non-linear spline representation of the effect of disease duration (Fig. 3). Therefore, our study provides an exhaustive analysis of retinal thickness change rates across MS patients of different subgroups.

The correlation of pRNFL, mRNFL, and GCIPL with VA and VEP latency in both relapsing and progressive subtypes and phases of the disease in our study indicates the functional relevance of these structural measurements and corroborates and expands upon previous reports[39–41].

In line with a previous study[14], pRNFL and mRNFL were predictive of relapses in MS and RRMS (Table 2). In contrast to previous studies[4,5,8,11,13,17,42], only mRNFL thickness was associated with future disability progression in PPMS (Table 3). Reasons for this could be the fact that we included a heterogeneous group of patients with different disease duration at baseline OCT (Table 1) and different intervals of

assessments in contrast to previous studies, which focused on patients with predominantly early MS. We decided to show the results for patients without ON separately (Tables 2–4)[4,5,8–20] as retinal atrophy in the absence of ON may be considered a more suitable surrogate for chronic neurodegeneration and predictor for progression while the atrophy after ON mainly results from the presence and the severity of the inflammatory insult at the optic nerve.

While several previous studies analyzed associations between baseline OCT and subsequent disability progression and/or disease activity[4,5,18,19], we also included longitudinal follow-up assessments adjusting for disease duration and time to assessment (Tables 2–4).

Interestingly, lower GCIPL thickness (Table 2) and mRNFL, GCIPL, and INL thickness of the lowest tertile (mRNFL ≤ 29 µm, GCIPL ≤ 62 µm, INL ≤ 33 µm) (Supplementary Table 9) were associated with increased risk for future MRI progression/activity in PPMS patients (Table 4) suggesting that retinal atrophy in PPMS is driven by inflammation (new

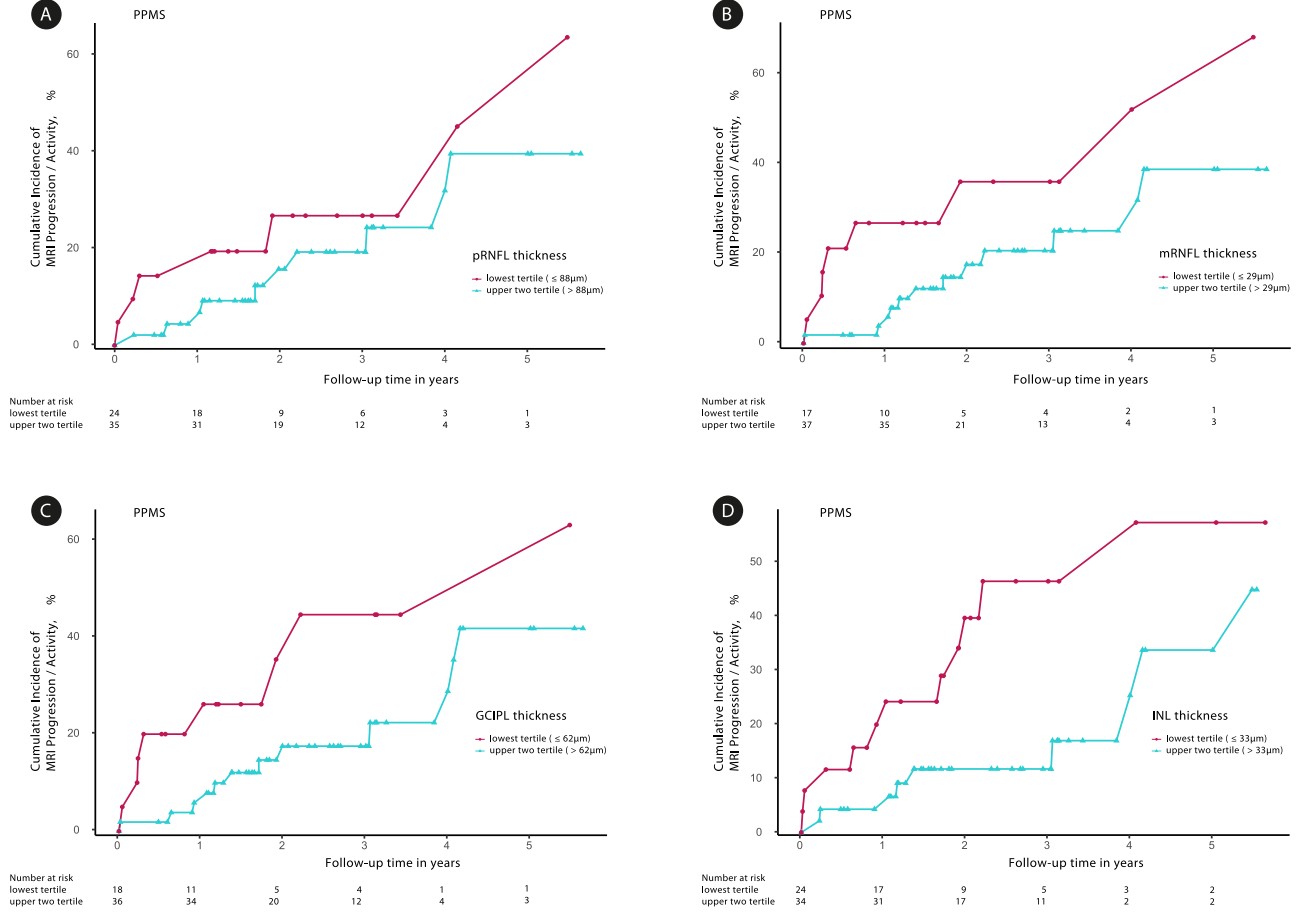

**Fig. 5 | Cumulative incidence of future MRI progression/activity in percent in PPMS patients according to pRNFL, mRNFL, GCIPL, and INL thickness at baseline.** The results of the Kaplan–Meier analysis are presented. Numbers at risk are the number of patients at risk of MRI progression/activity just before the selected timepoints. The difference in the number of patients between one timepoint and the next is the sum of the number of events and number of censored patients. **A** pRNFL peripapillary retinal nerve fiber layer; (**B**) mRNFL = macular retinal nerve fiber layer; (**C**) GCIPL = ganglion cell-inner plexiform layer; (**D**) INL = inner nuclear layer.

and enlarging T2-weighted/gadolinium-enhancing lesions) and has predictive value.

The cut-off values of pRNFL used for the Kaplan–Meier analyses (lowest tertile ≤ 88 μm) are in line with previous studies[5,11,13]. Segmentation of macular layers, in our study, was performed by using the 6-mm ETDRS grid which covers a larger area than in some previous studies[4,11]. This explains the lower cut-off values of GCIPL thickness in our study when compared to these studies[4,11].

Due to the heterogeneity of our clinical data with more active patients receiving high-efficacy disease-modifying therapies (DMTs), simple analyses of the effects of DTMs on retinal atrophy were not conclusive and more complex investigations for the different therapeutics were judged beyond the scope of this study. This issue will be addressed in a separate investigation. However, we considered a possible influence by adjusting prognostic models for DMTs (high-efficacy DMT, low-efficacy DMT, and no DMT) at the time of OCT assessment.

The main limitation of our study is the retrospective design without standardized follow-up intervals. Therefore, MRI data was not available for all visits and the intervals for capturing disability progression and relapse activity were very heterogeneous. The utilization of the EDSS for classification of the severity of disability of MS patients has known deficits due to its subjective character, inter-rater variability, its disregard of important sections as cognition, attention, and fatigue and low sensitivity to changes especially at higher scores, where scoring depends predominantly on ambulatory disability.

On the other hand, our study has several strengths, including the large cohort of 125 PPMS patients who so far have often been underrepresented in OCT studies. Further strengths are the long mean follow-up time, the segmentation of retinal layers, the clear differentiation of SPMS und PPMS patients, and the rigorous exclusion of ophthalmological pathologies with potential influence on structural and functional readouts.

Further studies over longer periods and with larger cohorts of patients and more standardized follow-up intervals are warranted to confirm our findings and to further evaluate the benefit of OCT measurements in predicting future disease activity and disability progression in SPMS and PPMS.

In summary, pRNFL and GCIPL were shown to be robust markers of neuroaxonal damage of the visual pathway throughout the disease course in both relapsing and progressive MS. However, longitudinal assessment of retinal thickness as currently performed in clinical routine, are subject to a considerable amount of measurement variability and may therefore not be suitable markers of progression on a single patient level. Further optimization of assessment tools, algorithms, and post-processing analyses are warranted to enhance the reliability and clinical utility of OCT measurements in MS.

## Methods
### Study approval
Patients and HCs participated in observational studies which were approved by the local ethics committees (Düsseldorf: 5794 R and

4389 R, Münster: 2017-754-f-S, Berlin: EA1/182/10 and EA1/163/12, Hamburg: PV4455, PV3961 and PV5557, Munich: 427-14), and provided written informed consent for participation.

## Study design and participants

This study was conducted and reported according to the STROBE (Strengthening the Reporting of Observational Studies in Epidemiology) guidelines. In this retrospective longitudinal cohort study, datasets of 590 patients with RRMS ($N = 261$), SPMS ($N = 139$), and PPMS ($N = 190$) who were diagnosed according to the revised McDonald criteria 2017[43] and 119 HCs were obtained from five MS centers (Düsseldorf, Münster, Berlin, Hamburg, and Munich). Patients had been recruited in the context of different non-interventional longitudinal OCT studies at the centers from 13.10.2009 to 18.05.2020. Inclusion criteria for this study were age ≥18 years and the availability of at least two consecutive OCT datasets with interval of >9 months. No statistical method was used to predetermine sample size. All patients were screened for glaucoma/ophthalmological diseases by medical history. A brief ophthalmological examination for exclusion of ophthalmological diseases was performed by trained specialists namely experienced neurologists and/or optometrists. Information on co-morbidities and abnormal laboratory parameters were available based on physician's letters, diagnostic findings, and blood analyses.

Exclusion criteria were any diseases of the optic nerve or retina not related to MS; a diagnosis of other neuroinflammatory disorders (i.e., neuromyelitis optica spectrum disorders); severe refraction anomalies ≥ ± 6 diopters; systemic conditions that could affect the visual system; treatment with substances with increased risk of iatrogenic retinopathy such as chemotherapy; insufficient scan quality according to the OSCAR-IB criteria[44,45] (Fig. 1). In MS patients, initial swelling and retinal atrophy in the context of acute ON has a major impact on retinal layer thickness[46]. For this reason, we excluded eyes with previous ON within 6 months to baseline OCT and those with ON between OCT measurements (Fig. 1).

## Procedures

**Clinical assessment.** All patients underwent physical, neurological, and OCT examinations at baseline and at each follow-up time point at one of the five participating MS centers. The following data were recorded: sex, date of birth, date of manifestation of patients' first symptoms, EDSS scores, episodes of ON, DMTs, occurrence of relapses, brain and spinal MRI progression/activity (new and enlarging T2-weighted/gadolinium-enhancing lesions).

EDSS scores were always assessed by the same team of specially trained neurologists at each participating center at the same visit as OCT. Researchers/technical assistants performing the OCT analysis were masked to EDSS results and those assessing disability by EDSS were masked at the time of the examination to OCT results. Disease duration was defined as time between the manifestation of first symptoms and the date of OCT examination. Disability worsening was defined as a documented increase in EDSS score compared to the previous measurement (≥1.0 point in case the EDSS score was <6.0, or ≥0.5 point if the EDSS score was ≥ 6.0) at a single time point. The EDSS increase did not have to be sustained. History of ON was assessed by the medical history and using the previously described OCT approach based on inter-eye GCIPL thickness difference of ≥4 µm[47].

## Visual acuity, visual evoked potentials, and OCT

Assessments of visual acuity and VEP were performed at the same visit as OCT. Standardized visual acuity measurement was performed in a subset of participants ($N_{RRMS} = 114$, $N_{SPMS} = 79$, $N_{PPMS} = 101$, $N_{HCs} = 59$) using retro-illuminated high-contrast Early Treatment of Diabetic Retinopathy Study charts (ETDRS) at 4 m. Patients used their habitual glasses/contact lenses when applicable.

Longitudinal VEPs were recorded in a subset of subjects ($N_{RRMS} = 94$, $N_{SPMS} = 40$, $N_{PPMS} = 50$, $N_{HCs} = 22$) and P100 peak-latencies were investigated. VEPs were performed using full-field monocular stimulation by pattern reversal black on white checkerboards following the local protocols at the centers (check size: 41' for University Hospital Düsseldorf and Münster | 60' for Charité-Universitätsmedizin Berlin) and in accordance with the International Society for Clinical Electrophysiology of Vision (ISCEV) standards with recording electrodes positioned at Oz (active) and Fz (reference)[48]. VEP recordings were repeated at least twice for each eye averaging >150 responses. Different VEP devices were used at the different centers, however, only assessments performed with the same device at baseline and follow-up time were considered. P100 latency was analyzed.

Spectral domain OCT examinations were performed with assessment of the pRNFL, mRNFL, GCIPL, and INL. OCT examinations were performed at each center using spectral domain OCT (SD-OCT, Spectralis, Heidelberg Engineering GmbH, Heidelberg, Germany) without pupil dilation at room light and with automatic real-time (ART) function for image averaging and an activated eye tracker. Automated retinal layer segmentation with manual correction of obvious segmentation errors was done by the Heidelberg Eye Explorer software version 1.9.10.0 (Heyex, Heidelberg Engineering, Heidelberg Germany). The pRNFL, mRNFL, GCIPL, and INL were reported in line with the APOSTEL 1.0 and 2.0 recommendations[49,50]. All segmentations were reviewed to confirm the accuracy of the segmentation, were manually corrected if necessary, and were evaluated for the presence of macular pathology (Table 1)[44,45]. Data on the pRNFL were obtained using circular ring scans with a 12° (-3.4 mm) diameter (1536 A-scans; 16 ≤ ART ≤ 99) placed around the optic nerve head. Data on the macular area were acquired using a macular volume scan (Münster, Düsseldorf, Berlin, Hamburg: 30° × 25° field, 61 vertical B-scans, 11 ≤ ART ≤ 18; Munich: 20° × 20°, 25 vertical B-scans, 21 ≤ ART ≤ 49) centered in the middle of the fovea. The mRNFL, GCIPL, and INL thickness were calculated within a 6 mm circle diameter around the fovea.

## Statistical analysis

All statistical analyses were performed using the best fit LMM identified by likelihood ratio tests using restricted maximum likelihood approach (SPSS Statistics 26.0 (IBM) or the lme4 package in R Studio (version 1.3.1093)). Results are presented as either mean with standard deviation (SD) or median with interquartile range (IQR) and $p < 0.05$ were considered significant. Due to the retrospective nature of the study no adjustment for multiple comparisons was made if it is not explicitly stated otherwise.

Chi-square-test was conducted to compare sex and the number of included eyes with ON, with clinical history of ON, with history of ON identified by cut-off of GCIPL inter-eye difference[47], and the number of patients on high-efficacy DMT and low-efficacy DMT between groups at baseline. Kruskal-Wallis-test was used to test for significant differences across disease subtypes at baseline regarding age, disease duration, EDSS, and follow-up times.

First, we investigated the effect of disease duration on retinal layer thickness, VEP latency, and VA at baseline by predicting the former based on a fixed effect for disease duration and random slope for person (Fig. 2). We calculated a separate model for each of the groups. To account for confounding influences, age and/or sex (Fig. 2) and age, sex, and/or ON (categorized in the categories (a) ON based on medical history, (b) ON based on inter-eye GCIPL thickness difference[47], and (c) no ON) (Supplementary Table 2; see Supplementary Information) were subsequently added to the model if the model fit could be improved. All models were compared based on likelihood-ratio-tests.

An interaction term with disease duration and disease course as well as participants' age, sex, and/or history of ON was included and its contrast-effects were checked using an F-test. If that test revealed

significant differences, estimated marginal means of linear trends were used to test differences in atrophy rates between disease courses for significance (using Tukey's method).

Patients participated in different observational studies at the five MS centers. Therefore, the disease duration at baseline spanned a large range for the different patient groups and follow-up assessments were performed at varying time points and with different frequencies. To account for the varying influence of disease duration at the different stages of the disease, the sample was divided into subgroups according to the disease duration for longitudinal analysis.

As we were specifically interested in progressive MS, the division of the HCs and RRMS group into different subgroups was orientated to the group of PPMS. The disease duration corresponded to the time since baseline OCT in HCs. In order to achieve an equal distribution of data within the different intervals the boundaries had to be set differently for SPMS than for RRMS and PPMS. The following intervals of disease duration were used: 0–3.5 years for HCs, 0–3.5 years, 3.6–5.5 years, 5.6–7.5 years, 7.6–10.5 years, 10.6–13.5 years, 13.6–16.5 years for PPMS and RRMS patients, additionally 16.6–20.5 years for PPMS, and 3.5–12.5 years, 12.6–16.5 years, 16.6–20.5 years, 20.6–25.5 years, 25.6–30.5 years, and over 30.6 years for SPMS patients (Fig. 3). Supplementary Table 6 (see Supplementary Information) displays the number of included eyes for each interval of disease duration by disease subtype.

In Fig. 3A–D, we calculated the annualized thickness change rates for the different intervals of disease duration (colored boxplots) for each disease type as the difference between two consecutive thickness measurements, divided by the time between the measurements in years. If more than two OCTs were in one interval, several annualized thickness change rates were entered for this subject in the interval. In cases with measurement intervals spanning several disease duration intervals the calculated annualized thickness change rate was considered for the interval of the follow-up scan. Further, we calculated the overall annualized thickness change rates of the different layers in μm (boxplots bordered in red) by calculating the average of all annualized thickness change rates of the different intervals of disease.

These binned analyses were only used to obtain the boxplots displaying medians and interquartile ranges in Fig. 3A–D for graphical representation. LMM analyses were used for hypothesis testing (calculation of the significances presented in Supplementary Tables 4 and 5). In order to obtain parameter estimates for the annualized thickness change rate for the different intervals of disease duration (Supplementary Table 5) and for the overall annualized thickness change rate over time (Supplementary Table 4), retinal layer thickness was predicted based on the main effect of disease duration with a random intercept per eye nested in subjects and a random slope per subject using a LMM. A random slope was only introduced for subject, as, naturally, disease duration did not vary between eyes but only between subjects. Therefore, no random slope for eye was included in the model. The annualized thickness change rate estimates were derived from the parameter estimate of the main effect of disease duration.

If this complex model resulted in a singular fit or nonconvergence of the model, it was reduced to a simpler model eliminating the least relevant part(s) of the overly complex model based on the smallest variance of the random effect. Subsequently, participants' age at assessment and sex were added as covariates and likelihood-ratio-tests were carried out to select the best model (Supplementary Tables 4 and 5).

In order to account for the high variability of the thickness measurements, we performed empirical Bayes estimates of BLUPs of the longitudinal effect of disease duration (subtracting the baseline disease duration for each patient from the disease duration at assessment) with time since baseline, age and sex as predictors, reporting histograms of estimates across disease courses (Supplementary Table 7) and eye wise OLS with disease duration as only predictor and aggregated estimated regression coefficients (Supplementary Table 8). To decrease the high variability associated with the small number of observations in each stratum of disease duration, coefficients were estimated over the entire disease duration. To test differences between the disease courses, we ran an ANOVA on empirical Bayes estimates of BLUPs of annualized thickness change rates (Supplementary Table 7), using post hoc tests (Tukey) to determine which groups differed with respect to their atrophy rates.

To further reduce the variability of OLS estimates, we discarded eyes with less than three OCTs (follow-up median (IQR): 3.95 years (2.97–6.03 years), Supplementary Fig. 5). To test the aggregated coefficients for significance, one-sample $t$-tests were run for each outcome and disease course. To test whether there are significant differences between disease courses, we performed an ANOVA with random intercepts per subject. In cases where the ANOVA revealed significant group differences, we conducted a Tukey post-hoc test to identify which groups differed significantly.

While overall fixed effect estimates (Supplementary Tables 4 and 5) capture both cross-sectional and longitudinal effects, described BLUPs (Supplementary Table 7) and eye wise OLS (Supplementary Table 8) capture exclusively longitudinal effects of disease duration and therefore provide an additional perspective.

To analyze longitudinal effects of disease duration on VEP latency and VA, no intervals for different periods of disease duration were created. The reasons were that we wanted to avoid power issues, and that we did not expect differential effects. Model computation followed the processes for longitudinal analysis of retinal layer thickness as described above. To model the influence of the retinal layer thickness on VA and VEP latency, VA and VEP latency (Fig. 4) were predicted based on a fixed effect and random slope for retinal layer thickness per eye nested in subject. Again, age and sex were added, and model comparison was based on likelihood-ratio-tests.

These models were part of the Gaussian family, calculated maximum likelihood based on Laplace approximation and modeled the relationship between predictors and outcome variables with an identity link function, and Gaussian errors. The random effects' a priori covariance matrices were unstructured, and collinearity was assessed by the variance inflation factor with a predetermined threshold of 5 as suggestive of multicollinearity.

As an alternative statistical approach to model cross-sectional and longitudinal effects of disease duration (and age) on retinal layers thickness, VEP latency, and VA, b-splines were fitted in a mixed model with random intercept for eyes and controlling for gender (Supplementary Figs. 1–3 (see Supplementary Information), Fig. 3). Knots were placed in equally spaced percentiles (e.g., knots were placed at the 25%, 50% and 75% percentiles when using three knots). The number of knots and polynomial degrees (both between 1 and 5) were chosen such that 10-fold cross validation mean squared error (MSE) was minimized. This was done for each disease course and dependent variable separately.

Prognostic models of EDSS progression, MRI progression/activity, and relapses (Tables 2–4) were calculated using binary logistic mixed effects models with eye-specific random intercepts. This approach allowed us to account for the hierarchical structure of our data with two eyes per subject, repeated OCT measurements within subjects, and individual differences in baseline values.

For the prediction of disease activity and disability progression (Tables 2–4) all OCT measurements except the last OCT were included in the logistic mixed effects models. For the prediction of MRI progression/activity, the logistic regressions considered each of four possible MRIs between one OCT and the next OCT as a separate observation. Therefore, a single OCT could yield up to four observations (this was the maximum number of MRI observations, which one of our patients had). For the prediction of relapses, we aggregated all

four possible relapses between one OCT and the next OCT. Thus, each OCT yielded a maximum of one observation for the analysis. Therefore, there are significantly fewer observations for relapses than for MRI. For the prediction of EDSS progression, the logistic regressions considered each EDSS (EDSS assessment was always performed with OCT measurement) after the considered OCT except the last one (since we cannot observe worsening after the last measurement). Thus, the analysis for EDSS considered one OCT less than for the analyses, resulting in a smaller sample size for EDSS.

We adjusted the analyses for participants' age at assessment, sex, EDSS at assessment, and DMT (high-efficacy DMT, low-efficacy DMT, and no DMT) if necessary, as indicated by a significant likelihood-ratio-test in favor of the more complex model. Moreover, logistic regressions were adjusted for the time to assessment of EDSS, MRI, and relapses. This was the interval between OCT and EDSS worsening for the prediction of EDSS, the interval between OCT and each MRI for the prediction of MRI progression/activity, the interval between OCT and relapse (if relapse occurred before next OCT) or the OCT inter-scan interval (if no relapse occurred) for the prediction of relapses.

All continuous covariates were standardized in order to yield a more balanced optimization problem for model fitting.

For the prediction of disease activity, we calculated the change rates as thickness changes (difference between OCT values) per time between measurements.

We performed Kaplan–Meier analysis with Cox proportional hazards models to assess the value of each OCT parameter at baseline as a potential risk factor for suffering disability worsening, MRI progression/activity, and relapses. The mean of the values of pRNFL, mRNFL, GCIPL, INL thickness for both eyes at baseline were used in the Kaplan–Meier analysis (Fig. 5) and Cox regression models (Supplementary Table 9). For the Cox model, only the time to first relapse, MRI activity or disability worsening (see methods) after baseline was analyzed. Recurrent events were not considered. For Kaplan-Meier and Cox regression analyses, we grouped patients into tertiles according to the thickness of pRNFL, mRNFL, GCIPL, or INL at the time of study enrolment. We compared the risks associated with being in the lowest tertile (pRNFL ≤ 88 μm, mRNFL ≤ 29 μm; GCIPL ≤ 62 μm; INL ≤ 33 μm) versus the two upper tertiles. The same stepwise feature selection as in the prediction analyses was used in the multiple Cox proportional hazard regression model. We report the hazard ratio (HR) with 95% CIs (Supplementary Table 9).

### Reporting summary

Further information on research design is available in the Nature Portfolio Reporting Summary linked to this article.

## Data availability

The datasets generated during and/or analyzed during the current study have been deposited in a publicly open data repository: https://doi.org/10.5281/zenodo.11106449[51]. Source data are provided with this paper.

## Code availability

All statistical codes for presented results are available under: https://github.com/AlexHartmann00/oct-ms. The codes are citable by obtaining a DOI for the Github repository: https://doi.org/10.5281/zenodo.11096799[52].

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

## Acknowledgements

We thank all subjects, who participated in this study. The authors thank Joachim Gerß of the Institute for Biometrics and Clinical Research of the University Muenster for his excellent consulting and supervising assistance with statistical analyses. We thank Felix Krämer of the Aug. Krämer Kornbrennerei in Dortmund, Christoph Schröer of the Labor B design office in Dortmund, and Patrick Maerker of Masslevel in Berlin for their support in creating the illustrations.

## Author contributions

J.K., M.W., A.H., V.K., Y.U., I.E., M.N.M., M.Z.; M.D., J.I., J.I.L., J.H., T.K., M.K., V.H., C.H., J.P.S., H.G.Z., F.C.O., M.R., A.U.B., F.P., O.A., H.P.H., H.W., S.G.M., P.A. contributed to the acquisition and quality control of clinical and afferent visual system data. C.B. and A.H. performed the data analysis. J.K., S.G.M., P.A. contributed to the conception and design of the study. J.K. and P.A. drafted the manuscript. All authors read and approved the final manuscript.

## Funding

## Competing interests

J.K. has received honoraria for lecturing from Biogen, Novartis, Sanofi Genzyme, Roche, Mylan and Teva, and financial research support from Sanofi Genzyme, Novartis, Roche, and Amicus Therapeutics. M.D. has received speaker honoraria from Merck. J.-I.L. has received honoraria for speaking/consultation from Boehringer Ingelheim, Allergan, Abbvie, Novartis, Ipsen, Teva, Lilly and Daiichi-Sankyo as well as travel grants from Bayer Healthcare, Merz, Allergan, Abbvie, Teva and Ipsen outside the submitted work. J.H. reports grants from the Friedrich-Baur-Stiftung, Merck and Horizon, personal fees and non-financial support from Alexion, Horizon, Roche, Merck, Novartis, Biogen, BMS and Janssen, and non-financial support from the Guthy-Jackson Charitable Foundation and The Sumaira Foundation. T.K. has received speaker honoraria and/or personal fees for advisory boards from Roche Pharma, Alexion/Astra Zeneca, Horizon, Merck, Chugai and Biogen. The institution she works for has received grant support for her research from Bayer-Schering AG, Novartis and Chugai Pharma in the past. None resulted in a conflict of interest. M.K. received compensation for serving on scientific advisory boards or speaker fees from Biogen Idec, Genzyme, MedDay Pharmaceuticals, Merck Serono, Novartis, Roche, Sanofi Aventis and Teva. He has received research support from Biogen Idec and Sanofi US. C.H. has received grants and speaker honaria from Merck, Novartis, Roche. J.-P.S.

has received research grants from Genzyme and honoraria for an advisory board from Alexion. H.G.Z. received research grants and speaking honoraria from Novartis. F.O. receives grants from the American Academy of Neurology, the National Multiple Sclerosis Society, and the Hertie foundation and board membership IMSVISUAL. M.R. received speaker honoraria from Novartis, Bayer Vital GmbH, Roche, Alexion, Horizon and Ipsen and travel reimbursement from Bayer Schering, Biogen Idec, Merz, Genzyme, Teva, Roche, Horizon and Merck, none related to this study. A.U.B. is cofounder and shareholder of startups Motognosis and Nocturne. He is named as inventor on several patent applications description MS serum biomarkers, perceptive visual computing, and retinal image analysis. F.P. serves as an Associate Editor for Neurology: Neuroimmunology & Neuroinflammation, reports research grants and speaker honoraria from Bayer, Teva, Genzyme, Merck, Novartis, MedImmune, Allmirall, Shire, Alexion, Chugai and is member of the steering committee of the OCTIMS study (Novartis). O.A. received grants from the German Research Foundation (DFG), Euge'ne Devic European Network (EU-FP7), German Ministry of Education and Research, and the Schaufler Foundation; honoraria for lectures from Almirall, Novartis, Bayer, Genzyme, Teva, Merck Serono, Biogen, Roche and Medimmune; and received travel/accommodation/meeting expenses from Novartis, Bayer Schering and Merck Serono. H.-P.H. received fees for serving on steering committees from Biogen Idec, GeNeuro, Sanofi Genzyme, Merck, Novartis Pharmaceuticals, Octapharma, Opexa Therapeutics, Teva Pharmaceuticals, MedImmune, Bayer HealthCare, Forward Pharma, and Roche; fees for serving on advisory boards from Biogen Idec, Sanofi Genzyme, Merck, Novartis Pharmaceuticals, Octapharma, Opexa Therapeutics, Teva Pharmaceuticals, and Roche; and lecture fees from Biogen Idec, Sanofi Genzyme, Merck, Novartis Pharmaceuticals, Octapharma, Opexa Therapeutics, Teva Pharmaceuticals, MedImmune, and Roche. H.W. received compensation for serving on Scientific Advisory Boards/Steering Committees for Bayer Healthcare, Biogen Idec, Sanofi Genzyme, Merck Serono, and Novartis. He has received speaker honoraria and travel support from Bayer Vital GmbH, Bayer Schering AG, Biogen, CSL Behring, EMD Serono, Fresenius Medical Care, Genzyme, Merck Serono, Omniamed, Novartis, and Sanofi Aventis. He has received compensation as a consultant from Biogen Idec, Merck Serono, Novartis, Roche, and Sanofi Genzyme. H.W. also received research support from Bayer Healthcare, Bayer Vital, Biogen Idec, Merck Serono, Novartis, Sanofi Genzyme, Sanofi US, and Teva. S.G.M. received honoraria for lecturing and travel expenses for attending meetings from Almirall, Amicus Therapeutics Germany, Bayer Health Care, Biogen, Celgene, Diamed, Genzyme, MedDay Pharmaceuticals, Merck Serono, Novartis, Novo Nordisk, ONO Pharma, Roche, Sanofi-Aventis, Chugai Pharma, QuintilesIMS, and Teva. His research is funded by the German Ministry for Education and Research (BMBF), Deutsche Forschungsgemeinschaft (DFG), Else Kröner Fresenius Foundation, German Academic Exchange Service, Hertie Foundation, Interdisciplinary Center for Clinical Studies (IZKF) Muenster, German Foundation Neurology, and by Almirall, Amicus Therapeutics Germany, Biogen, Diamed, Fresenius Medical Care, Genzyme, Merck Serono, Novartis, ONO Pharma, Roche, and Teva. P.A. received honoraria for lecturing and travel expenses for attending meetings from Abbvie, Biogen, Celgene, Genzyme, Hexal, Ipsen, Lilly, Merck Serono, Merz Pharmaceuticals, Novartis, Pfizer, Roche, Sanofi-Aventis, and Teva. His research is funded by Deutsche Forschungsgemeinschaft (DFG), EFRE-NRW, and by Abbvie, Biogen, Celgene, Merck Serono, Merz Pharmaceuticals, Novartis, and Roche. C.B., M.W., A.H., V.K., Y.U., I.E., M.N.-M., M.Z., J.I., V.H. report no competing interests.

## Additional information

¹Department of Neurology with Institute of Translational Neurology, University Hospital Münster, Münster, Germany. ²Department of Neurology, Medical Faculty and University Hospital Düsseldorf, Heinrich-Heine-University Düsseldorf, Düsseldorf, Germany. ³Department of Neurology, Kliniken Maria Hilf, Mönchengladbach, Germany. ⁴Hanseatic Statistics, Hamburg, Germany. ⁵Institute of Clinical Neuroimmunology, LMU Hospital, Ludwig-Maximilians University München, München, Germany. ⁶Biomedical Center, Faculty of Medicine, Ludwig-Maximilians University München, München, Germany. ⁷Munich Cluster for Systems Neurology (SyNergy), Munich, Germany. ⁸Institute of Neuroimmunology and Multiple Sclerosis (INIMS), University Hospital Hamburg-Eppendorf, Hamburg, Germany. ⁹Department of Neurology, University Medical Centre Hamburg-Eppendorf, Hamburg, Germany. ¹⁰Aix-Marseille University, CNRS-CRMBM, UMR, 7339 Marseille, France. ¹¹APHM La Timone, CEMEREM, Marseille, France. ¹²Experimental and Clinical Research Center, Max-Delbrück Center for Molecular Medicine and Charité - Universitätsmedizin Berlin, corporate member of Freie Universität Berlin and Humboldt-Universität zu Berlin, Berlin, Germany. ¹³Department of Neurology, Center for Neurology and Neuropsychiatry, LVR-Klinikum, Heinrich-Heine-University Düsseldorf, Düsseldorf, Germany. ¹⁴Brain and Mind Center, University of Sydney, Sydney, NSW, Australia. ¹⁵Department of Neurology, Palacky University Olomouc, Olomouc, Czech Republic. ¹⁶These authors contributed equally: Sven G. Meuth, Philipp Albrecht. ✉e-mail: julia.kraemer@ukmuenster.de; phil.albrecht@gmail.com

