## [Peer Review File · Nature Communications]

Evolution of retinal degeneration and prediction of disease activity in relapsing and progressive multiple sclerosisREVIEWER COMMENTS

Reviewer #1 (Remarks to the Author):

In this study, Kramer et al report results of a retrospective longitudinal analysis of a large cohort of people with MS monitored with serial OCT, VA and/or VEP. They report results from a number of analyses, including associations including of measures derived from the above methods with disease duration at baseline, their change over time, and associations with disease activity during follow-up, stratified by disease subtype (RRMS, PPMS, SPMS) and disease duration. The topic is clearly of interest and the study is rigorous, reported in accordance with existing guidelines, by a study team with extensive expertise and experience in conducting studies involving visual pathway outcome measures in MS. I have a few suggestions/comments which are listed below:

1) The approach to the analysis focuses heavily on “disease duration”, which is defined based on symptom onset. However, it is fairly clear given knowledge regarding RIS and prodromal MS symptoms that symptom duration does not adequately capture the duration of the disease, and that subclinical changes (including retinal atrophy) may be present at the earliest stages of the disease, including in RIS. In clinical practice, defining symptom onset can often be challenging in the setting of retrospective ascertainment from the patient history. Additionally, the concept of disease duration differs between RRMS/SPMS and PPMS, given the different definitions, further complicating the interpretation of this variable. Similarly, the distinction of PPMS and SPMS can may be fairly arbitrary (e.g. if a patient with PPMS and history of ON as defined by inter-eye asymmetry had reported visual symptoms c/w ON they would be classified as SPMS) and current thinking is that PPMS/SPMS represent similar underlying disease processes rather than distinct entities. Given the issues outlined above, I feel that it would be quite informative to perform sensitivity analyses in order to assess the robustness of the findings reported in the manuscript, including the following: a) combined analyses of PPMS/SPMS with disease duration defined as time from transition to progressive MS for SPMS and time from symptom onset for PPMS (essentially putting them on the same time scale b) reporting the results by age (ideally modeled flexibly as a continuous variable [see second comment below] or as subgroups) for both MS and HC

2) Please report the sample sizes for each disease duration subgroup by disease subtype. It is difficult to interpret these findings given that each bin is quite small (e.g. there are 87 SPMS participants divided across 6 disease duration subgroups, so even if divided equally each group’s sample size would be $87/6 \approx 14$). The precision of the estimates obtained in each bin is low, given the small sample sizes, which renders the interpretation of the findings quite challenging, and we can see in Supplementary Table 3 that the estimates bounce around quite a bit, likely because of this issue, rather than a true underlying biological difference. I would recommend considering modeling disease duration flexibly using for example polynomial regression or spline functions (e.g. see PMID: 29971334).

3) It is simply stated that “the following analyses were corrected for participants’ age and sex”, but otherwise no information regarding these features and their associations with retinal layer atrophy is reported. See comment 1 above regarding reporting analyses for age. Furthermore, the authors should

clarify whether baseline age or time-varying age was included in the longitudinal models. If time-varying age was included, the model interpretation presented for the rates of atrophy that are presumably derived from inclusion of follow-up time at each OCT timepoint will not be interpretable, since both age and time will be increasing by the same amount at each visit.

4) Please report 95% CI for the beta coefficients in Supplementary Tables 2 and 3, in order to inform regarding the precision of the estimates (currently only p-values are reported).

5) Analyses do not appear to have been adjusted for history of optic neuritis, which is one of the strongest factors associated with retinal layer thickness, and the prevalence of which is expected to vary by disease duration and subtype (higher prevalence of ON with longer disease duration and no history of ON [except if defined by OCT] in PPMS). I recommend that the analyses be adjusted for history of ON, especially the baseline analyses.

6) It seems that ON was defined using not only clinical history of ON, but also based on inter-eye GCIPL thickness differences of $\geq 4\mu\text{m}$. While this has been proposed as a cut-off for determining a history of ON, it is also possible that the presence of subclinical optic neuropathy without a history of clinical ON may reflect disease activity/severity and propensity to neurodegeneration, and be informative in regards to prediction of disability. It would be useful to clarify in Table 1 what proportion of eyes with history of ON was determined by clinical history vs OCT criteria.

7) Disease duration was not considered for analyses of VEP and VA, but this is not clearly justified. The authors report that they wanted to avoid power issues given the lower sample sizes for these cohorts, but also report that they did not expect differential effects. While I agree with the former, this could be addressed by modeling disease duration as a continuous variable (as previously suggested, including flexible approaches, which would be a valid analysis and avoid the issues with creating small bins of participants, which notably are problematic even for the OCT analyses, as pointed out previously in this review). Also, it is not clear why differential effects would not be expected on VEP latencies, since this measure represents overlapping underlying disease processes with OCT.

8) The approach to the predictive analyses of disease activity/outcomes is unclear and needs to be described in greater detail. It is reported that the authors considered OCT measurements where information on these outcomes was available at the time point of the subsequent OCT. A few questions arise in this setting: a) Were there participants with more than one inter-scan interval included in the analysis b) How much did the inter-scan interval vary between those included? The current approach of using logistic regression does not account for the time, and someone with a longer time would be at a greater risk of relapse, EDSS progression or MRI activity, purely on the basis of having a longer time c) How were the longitudinally assessed retinal layer thickness change rates derived for inclusion in the predictive model analyses?

9) In the methods it is stated that disability worsening was defined as “documented increase in EDSS score as a consequence of relapses or disease progression”. This does not appear based on the description in the methods to represent “confirmed” or “sustained” disability worsening (i.e. confirmed on two separate examination 3 or 6 months apart, which is a typical definition used in clinical

trials). However, in the results section the term “confirmed EDSS worsening” is used. This needs to be clarified and reconciled between the methods and results sections.

10) In the introduction it is stated that “longitudinal investigations of retinal morphology ... and a plateau effect with a longer disease duration”. Only a single longitudinal study of 135 participants is cited here to support this rather strong claim that the authors are making. I recommend adding supporting literature for this or softening the claim. Notably, in the prior analysis by Sotirchos et al, higher disease duration (when accounting for factors including disease subtype, age and disability) was associated with slower pRNFL thinning, but not GCIPL thinning. Furthermore, it is stated by the authors in the introduction that in that study atrophy rates were not analyzed separately for PPMS and SPMS, however they are reported separately in the manuscript and were similar in PPMS and SPMS.

11) Table 1. There seems to be an error, as “All MS subjects” are reported as 505, while the sum of the RRMS, PPMS and SPMS groups is 407. Also, I suggest that the authors consider softening their strong priority claims regarding the size of the study in the conclusions (first paragraph of discussion), since the sample size difference from some prior studies with overlapping aims/analyses (e.g. Sotirchos et al. *Ann Neurol*: 364 MS participants, Martinez-Lapiscina et al. *Lancet Neurol* 2016: 879 MS participants) is rather trivial (e.g., refer to the following editorial regarding this topic, PMID: 33603198, from the editors of a Nature publishing group journal).

Reviewer #2 (Remarks to the Author):

This is a very interesting study, including a large cohort of MS patients of different subtypes and HC. The main messages are the uniform decrease of thickness of both pRNFL and GCIPL in RRMS and progressive MS and the prognostic value of pRNFL, mRNFL and GCIPL thickness in regards to EDSS, relapses and MRI activity in the entire cohort. The study has several strengths, such as inclusion of patients with SPMS and PPMS, robust stats, and longitudinal assessments.

Few points to consider:

1. The term "MRI activity" can be misleading as in the current study refers to both gad-enhancing lesions as well as new lesions. Please re-phrase; may use MRI progression/activity.
2. Please clarify if the SPMS patients have active or non-active SPMS. An EDSS score of 5.5 (1.5-8) seems somewhat low for a mean duration of 20.4 years (4-43.5) Please give more details for this group.
3. Were all the assessments, including VA, EDSS, VEP and MRI occurred at the same visit? Please clarify.
4. Were the patients screened for glaucoma/ophthalmological diseases by an eye specialist? Were the patients screened for hemoglobin A1C or their co-morbidities were self-reported? Please add/clarify.
5. Please explain the logic behind dividing the disease duration into certain intervals: 0-3.5, 3.6-5.5, 5.6-7.5 etc, for RRMS and PPMS; similarly the 3.5-12.5 y, etc for SPMS.
6. Table 1. The numbers do not add up for the MS patients (195+ 125+87=407; not 505; please revise.

Reviewer #3 (Remarks to the Author):

Kraemer et al conducted a longitudinal study analyzing changes of retinal layer thickness and visual function over time and their predictive value for subsequent disease activity and disability progression in a relatively large cohort of RRMS, SPMS, and PPMS patients. The objective of identifying OCT measures that predict patients at high risk of disease activity/progression is extremely relevant. Results show that pRNFL and GCIPL are robust markers of neuroaxonal damage of the visual pathway throughout the disease course in both relapsing and progressive MS and that SD-OCT assessment performed at a single time point can predict future disease activity and disability progression.

Some quite unique aspects for this study are: (I) the high proportion of progressive patients, (II) the fact that SPMS and PPMS were considered separately and (III) the large number of follow-up assessments (leading to the observation that longitudinal assessments of retinal thickness are not suitable on a single patient level). Overall, the study is relevant and well conducted, although largely confirmatory of prior studies.

The following issues should be addressed:

- The authors state that 2651 measurements of 195 RRMS, 87 SPMS, 125 PPMS patients, and 98 HCs were included in the analysis; however, it should be specified how many OCT/MRI were available for each timepoint and how many patients had 2, 3 or more timepoints; this information could be added to the supplementary material or specified in Fig. 1
- Were spinal cord lesions considered when exploring disease activity or just brain? If only data regarding brain MRI activity were available, it should be listed as a limitation and discussed.
- As reported in Table 1, percentage of patients on DMTs at baseline differed between groups (as expected, the percentage of treated patients was higher in RRMS as compared to the progressive groups of patients). Although authors state that investigations for the different therapeutics were judged beyond the scope of this study, we think that analyses should somehow take into consideration whether patients were treated or not (adding it as a covariate in the predictive models? sensitivity analyses?). Indeed, not only treatment strongly influences disease activity (which is one of the main outcomes explored in this study), but it has also been suggested that DMT initiation may prevent GCIPL thinning and may lead to reduction in INL thickness (Bsteh G, et al. *Eur J Neurol.* 2021, ref 7; Knier et al. *Brain* 2016, ref 13). If available, the duration of treatment for the MS patients on their DMT at the time of the OCT should be also reported. Moreover, it should be stated whether or not all patients remained in the same DMT during the follow-up and eventually those who did not should be analyzed differently.
- One relevant finding of this study is that pRNFL, macular RNFL, and ganglion GCIPL thickness predicted future disease activity. However, this correlation was evident when patients with RRMS, SPMS and PPMS were included in the analysis all together, while analyzing the MS subtypes separately revealed no significant prediction. According to the authors' discussion, it could be due to insufficient sample size for the subgroups. On this matter, due to the retrospective nature of the study including patients from 5 different german centers, authors decided to group MS individuals into subgroups with variable disease duration intervals to achieve an equal distribution of data among cohorts introducing, in my view an arbitrary bias in the analysis. It should be considered however that other papers with a similar sample size found an association between OCT metrics and disease activity/progression (Bsteh G, et al. *Eur J Neurol.* 2021, ref 7; Cellerino et al, *J Neuroophthalmol.* 2021). In the Cellerino manuscript the association between OCT metrics and disability worsening was statistically significant in RRMS but not in

PMS patients. Similarly, Martinez-Lapiscina et al suggested that a single pRNFL assessment could be predictive of disability worsening in MS, but they did not detect significant differences in baseline OCT features between patients with PMS whose disability worsened and those who remained stable (GCIPL data were not available at the time). In the aforementioned studies, the weak correlation between baseline OCT metrics and subsequent disability in patients was explained, at least partly, by the disproportioned retinal injury resulting from higher age, disease duration and baseline disability, which characterize PMS patients. In this paper, Kraemer et al deeply explore the association with disease duration, but never take into account baseline disability (which – according to Table 1 – ranges from 0 to 7 in the RRMS group). The possible influence of baseline disability in predicting clinical outcome over time should be explored (maybe excluding from the RRMS group patients with EDSS>5.5, scores which are atypical for a relapsing-remitting disease course). Altogether, these results should be further discussed in the context of previous findings.

- When authors explore the association with disability worsening, it should be specified whether EDSS progression was assessed at a single time-point or confirmed (i.e., EDSS assessment repeated at 3 or 6 months to confirm if disability progression was “sustained”).

Minor points

- Were there additional exclusion criteria that were not mentioned in this manuscript?

Prior/concomitant administration of certain chemotherapeutic agents should be considered as exclusionary criteria (as it actually seems to be the case when looking at Fig. 1); however, iatrogenic optic neuropathy is not mentioned in the text.

- In the methods section, the authors define the study “exploratory”; accordingly, they did not adjust for multiple comparison. Since several papers have already explored changes in terms of OCT metrics over time and their predictive value in MS (even if with a lower percentage of progressive patients), authors should avoid defining this study “exploratory”.

Reviewer #1

In this study, Kramer et al report results of a retrospective longitudinal analysis of a large cohort of people with MS monitored with serial OCT, VA and/or VEP. They report results from a number of analyses, including associations including of measures derived from the above methods with disease duration at baseline, their change over time, and associations with disease activity during follow-up, stratified by disease subtype (RRMS, PPMS, SPMS) and disease duration. The topic is clearly of interest and the study is rigorous, reported in accordance with existing guidelines, by a study team with extensive expertise and experience in conducting studies involving visual pathway outcome measures in MS. I have a few suggestions/comments which are listed below:

R1.1: The approach to the analysis focuses heavily on “disease duration”, which is defined based on symptom onset. However, it is fairly clear given knowledge regarding RIS and prodromal MS symptoms that symptom duration does not adequately capture the duration of the disease, and that subclinical changes (including retinal atrophy) may be present at the earliest stages of the disease, including in RIS. In clinical practice, defining symptom onset can often be challenging in the setting of retrospective ascertainment from the patient history. Additionally, the concept of disease duration differs between RRMS/SPMS and PPMS, given the different definitions, further complicating the interpretation of this variable. Similarly, the distinction of PPMS and SPMS can may be fairly arbitrary (e.g. if a patient with PPMS and history of ON as defined by inter-eye asymmetry had reported visual symptoms c/w ON they would be classified as SPMS) and current thinking is that PPMS/SPMS represent similar underlying disease processes rather than distinct entities.

Given the issues outlined above, I feel that it would be quite informative to perform sensitivity analyses in order to assess the robustness of the findings reported in the manuscript, including the following: a) combined analyses of PPMS/SPMS with disease duration defined as time from transition to progressive MS for SPMS and time from symptom onset for PPMS (essentially putting them on the same time scale b) reporting the results by age (ideally modeled flexibly as a continuous variable [see second comment below] or as subgroups) for both MS and HC.

A1.1: We thank the reviewer for this important comment. The time from transition to progressive multiple sclerosis (MS) is subject to debate and not clearly defined ¹. The main challenge is attributed to the growingly accepted concept that relapsing-remitting and secondary progressive MS (RRMS, SPMS) are a continuum of the same disease rather than two distinct entities ². Because the time from transition was not homogeneously assessed across the centers and not always documented in the databases of the different centers, we could not include this parameter in the analyses.

As proposed, we additionally performed the cross-sectional and longitudinal associative analyses of retinal layers in relation to age instead of disease duration in patients with RRMS, SPMS, primary progressive MS (PPMS), and healthy controls (HCs) (see **Supplementary Figure 2 and 3**). We added the following sentences to the results: “In addition, we have plotted the retinal layer thickness, visual acuity, and VEP latency at baseline depending on age (**Supplementary Figure 2**). Especially pRNFL and GCIPL thickness slowly decreased with increasing age in all MS subtypes (**Supplementary Figure 2**)” “For reasons of completeness, we also plotted longitudinal changes of retinal layer thickness, visual acuity, and VEP latency over time depending on age for the different

subgroups (**Supplementary Figure 3**). However, disease duration seems to be a much more significant factor for retinal atrophy over time than age.”

For reasons of completeness, we added **Supplementary Figure 2 and 3** to the supplementary material.

R1.2: Please report the sample sizes for each disease duration subgroup by disease subtype (FIGURE 3).

It is difficult to interpret these findings given that each bin is quite small (e.g. there are 87 SPMS participants divided across 6 disease duration subgroups, so even if divided equally each group’s sample size would be $87/6 \approx 14$). The precision of the estimates obtained in each bin is low, given the small sample sizes, which renders the interpretation of the findings quite challenging, and we can see in Supplementary Table 3 that the estimates bounce around quite a bit, likely because of this issue, rather than a true underlying biological difference. I would recommend considering modeling disease duration flexibly using for example polynomial regression or spline functions (e.g. see PMID: 29971334).

A1.2: We thank the reviewer for this very helpful comment. We adapted the manuscript accordingly and now report the number of included eyes for each interval of disease duration by disease subtype in the **Supplementary Table 5**. Moreover, we plotted longitudinal changes of retinal layer thickness, visual acuity, and VEP latency over time depending on disease duration for RRMS, SPMS, PPMS, and HCs as spline fits (see **new Figure 3**). We added the following sentence to the results: “In line with the results of the mixed linear regression models, spline fits demonstrated decrease of pRNFL, mRNFL and even more GCIPL in the different MS subgroups (**Figure 3A-C**).”

R1.3: It is simply stated that “the following analyses were corrected for participants’ age and sex”, but otherwise no information regarding these features and their associations with retinal layer atrophy is reported. See comment 1 above regarding reporting analyses for age.

Furthermore, the authors should clarify whether baseline age or time-varying age was included in the longitudinal models. If time-varying age was included, the model interpretation presented for the rates of atrophy that are presumably derived from inclusion of follow-up time at each OCT timepoint will not be interpretable, since both age and time will be increasing by the same amount at each visit.

A1.3: We thank the reviewer for this comment. Due to the number of subgroups and number of results reported in figures and tables, we chose to focus only on predictors of primary interest and to not report the results for all covariates for reasons of clarity. Participants’ age and sex are only control variables, for which all analyses were corrected. However, their coefficients were not relevant.

We agree that age and time increase by the same amount in each patient. However, each patient developed disease at different ages and multicollinearity analyses ($VIF < 5$) revealed that there was no substantial correlation. We therefore conclude that the coefficients can be interpreted. For clarification, we have specified this in the methods: “Subsequently, participants’ age at assessment and sex were added as covariates and likelihood-ratio-tests were carried out to select the best model”.

R1.4: Please report 95% CI for the beta coefficients in Supplementary Tables 2 and 3, in order to inform regarding the precision of the estimates (currently only p-values are reported).

A1.4: We thank the reviewer for this comment. We now added the 95% CI for the beta coefficients in **Supplementary Tables 3 and 4** (previous **Supplementary Tables 2 and 3**). We had to correct some p values in **Supplementary Table 4** because they resulted of the model without covariates. However, the changes were only marginal and did not change the significances. Only the group of HCs did not longer show GCIPL thickness loss over a mean follow-up of 3.5 years while the group of SPMS demonstrated mRNFL atrophy in the later phases. We changed the results and discussion accordingly: “mRNFL atrophy was observed in the early phases of disease in RRMS (<3.5 years) and in the later stages in RRMS and PPMS (>10.6 years) and SPMS (>25.6 years) (**Figure 3B**)” “Besides retinal atrophy in all MS subtypes at different phases of disease, the group of HCs also showed pRNFL, and mRNFL thickness loss over a mean follow-up of 3.5 years. The amount of retinal atrophy observed in our HCs (Figure 3A-D) is in line with previous studies ^{19,20,22}. A recent study demonstrated that with increasing age, the rate of pRNFL atrophy in MS approaches rates similar to those expected with normal aging ²⁰, which is in line with our findings.”

R1.5: Analyses do not appear to have been adjusted for history of optic neuritis, which is one of the strongest factors associated with retinal layer thickness, and the prevalence of which is expected to vary by disease duration and subtype (higher prevalence of ON with longer disease duration and no history of ON [except if defined by OCT] in PPMS). I recommend that the analyses be adjusted for history of ON, especially the baseline analyses.

A1.5: We thank the reviewer for this important comment. We have now included optic neuritis as a covariate in our baseline analyses (cross-sectional associative analyses of retinal layer thickness, visual acuity, and VEP latency in relation to disease duration) and in our prediction analyses and added the following sentences to the methods: “Moreover we adjusted the model for ON (categorized in the categories a) ON based on medical history, b) ON based on inter-eye GCIPL thickness difference ⁴⁰, and c) no ON).” and “Prognostic models of EDSS progression, MRI progression/activity, and relapses (**Table 2**) were calculated using binary logistic mixed effects models with eye-specific intercepts. This approach allowed us to account for the hierarchical structure of our data with two eyes per subject, repeated OCT measurements within subjects, and individual differences in baseline values. We adjusted the analyses for disease duration, age and ON (categorized in the categories a) ON based on medical history, b) ON based on inter-eye GCIPL thickness difference ⁴³, and c) no ON), sex, baseline EDSS, DMT (high-efficacy DMT, low-efficacy DMT, and no DMT) if necessary, as indicated by a significant likelihood-ratio-test in favor of the more complex model.”

We added the following sentences to the results: “Adjusting the linear regressions between retinal layer thickness, VA, and VEP latency at baseline and disease duration for ON revealed the same findings for VA, VEP latency, and pRNFL thickness (**Supplementary Table 2**). The mRNFL and GCIPL thickness were additionally inversely associated with the disease duration in PPMS. Associations between the INL thickness and disease duration were not found in SPMS but in PPMS (**Supplementary Table 2**).”

For our longitudinal analyses, we chose to not include ON as a covariate. Based on the work by Balk and colleagues ³ we think that a previous ON will not have an influence on change

rates of retinal thickness because we explicitly excluded eyes with previous ON within 6 months to baseline OCT and those with ON between OCT measurements.

R1.6: It seems that ON was defined using not only clinical history of ON, but also based on inter-eye GCIPL thickness differences of $\geq 4\mu\text{m}$. While this has been proposed as a cut-off for determining a history of ON, it is also possible that the presence of subclinical optic neuropathy without a history of clinical ON may reflect disease activity/severity and propensity to neurodegeneration, and be informative in regards to prediction of disability. It would be useful to clarify in Table 1 what proportion of eyes with history of ON was determined by clinical history vs OCT criteria.

A1.6: We thank the reviewer for this comment. We have now specified the proportion of eyes with clinical history of ON vs. ON identified by cut-off of GCIPL inter-eye difference in Table 1.

R1.7: Disease duration was not considered for analyses of VEP and VA, but this is not clearly justified. The authors report that they wanted to avoid power issues given the lower sample sizes for these cohorts, but also report that they did not expect differential effects. While I agree with the former, this could be addressed by modeling disease duration as a continuous variable (as previously suggested, including flexible approaches, which would be a valid analysis and avoid the issues with creating small bins of participants, which notably are problematic even for the OCT analyses, as pointed out previously in this review). Also, it is not clear why differential effects would not be expected on VEP latencies, since this measure represents overlapping underlying disease processes with OCT.

A1.7: We thank for this important comment and addressed this point by modeling disease duration as a continuous variable using spline fits (see **new Figure 3**). We deleted the part “did not expect differential effects” from the sentence: “Because we wanted to avoid power issues, no intervals for different periods of disease duration were created.”

R1.8: The approach to the predictive analyses of disease activity/outcomes is unclear and needs to be described in greater detail. It is reported that the authors considered OCT measurements where information on these outcomes was available at the time point of the subsequent OCT. A few questions arise in this setting: a) Were there participants with more than one inter-scan interval included in the analysis, b) How much did the inter-scan interval vary between those included? The current approach of using logistic regression does not account for the time, and someone with a longer time would be at a greater risk of relapse, EDSS progression or MRI activity, purely on the basis of having a longer time a risk, c) How were the longitudinally assessed retinal layer thickness change rates derived for inclusion in the predictive model analyses?

A1.8: We thank the reviewer for this very important point.

a) yes; Repeated OCT measurements within single patients were considered for the prognostic models of EDSS progression, MRI progression/activity, and relapses adjusting for within subject effects. We added the following sentence to the methods: “Prognostic models of EDSS progression, MRI progression/activity, and relapses (**Table 2**) were calculated using binary logistic mixed effects models with eye-specific intercepts. This approach allowed us to

account for the hierarchical structure of our data with two eyes per subject, repeated OCT measurements within subjects, and individual differences in baseline values.”

b) the median (IQR) inter-scan interval is described in the legend of Table 2: “Mean time (SD) to EDSS assessment from each OCT to the next OCT: 1.83y (\pm 1.53y); mean time (SD) to MRI assessment from OCT to post-OCT MRI: 1.50y (\pm 1.30y); mean time (SD) to relapse assessment from OCT to post-OCT relapse: 1.70y (\pm 1.44y).”

Moreover, we revised our prediction analyses by adjusting for the time to assessment of EDSS, MRI, and relapses as now stated in the methods: “Moreover, logistic regressions were adjusted for the time to assessment of EDSS, MRI, and relapses. This was the OCT inter-scan interval for the prediction of EDSS, the interval between OCT and each MRI for the prediction of MRI progression/activity, the interval between OCT and relapse (if relapse occurred before next OCT) or the OCT inter-scan interval (if no relapse occurred) for the prediction of relapses.” This has changed the results of our prediction analysis, which had been impacted by differences in the time to assessment. We are very grateful for this reviewers comment and believe that the new analysis is much more valid.

c) We added the following sentence to the methods: “For the prediction of disease activity, we calculated the change rates as thickness changes (difference between OCT values) per time between measurements.”

R1.9: In the methods it is stated that disability worsening was defined as “documented increase in EDSS score as a consequence of relapses or disease progression”. This does not appear based on the description in the methods to represent “confirmed” or “sustained” disability worsening (i.e. confirmed on two separate examination 3 or 6 months apart, which is a typical definition used in clinical trials). However, in the results section the term “confirmed EDSS worsening” is used. This needs to be clarified and reconciled between the methods and results sections.

A1.9: We thank the reviewer for this important comment. We agree that the word “confirmed” or “sustained” disability worsening was misleading and not correct as we also considered EDSS progression without confirmation. We have corrected this in the manuscript removing the word confirmed.

R1.10: In the introduction it is stated that “longitudinal investigations of retinal morphology ... and a plateau effect with a longer disease duration”. Only a single longitudinal study of 135 participants is cited here to support this rather strong claim that the authors are making. I recommend adding supporting literature for this or softening the claim. Notably, in the prior analysis by Sotirchos et al, higher disease duration (when accounting for factors including disease subtype, age and disability) was associated with slower pRNFL thinning, but not GCIPL thinning. Furthermore, it is stated by the authors in the introduction that in that study atrophy rates were not analyzed separately for PPMS and SPMS, however they are reported separately in the manuscript and were similar in PPMS and SPMS.

A1.10: We thank you for this comment. We tapered down our statement in the introduction and added supporting literature: “Some longitudinal investigations of retinal morphology in mixed relapsing and progressive MS cohorts have suggested an attenuated atrophy rate of the inner retinal layers with a longer disease duration”. Additionally, we deleted the sentence

„However, in this study atrophy rates were not separately analyzed in primary and secondary progressive MS (PPMS, SPMS).“

R1.11: Table 1. There seems to be an error, as “All MS subjects” are reported as 505, while the sum of the RRMS, PPMS and SPMS groups is 407. Also, I suggest that the authors consider softening their strong priority claims regarding the size of the study in the conclusions (first paragraph of discussion), since the sample size difference from some prior studies with overlapping aims/analyses (e.g. Sotirchos et al. *Ann Neurol*: 364 MS participants, Martinez-Lapiscina et al. *Lancet Neurol* 2016: 879 MS participants) is rather trivial (e.g., refer to the following editorial regarding this topic, PMID: 33603198, from the editors of a Nature publishing group journal).

A1.11: We thank the reviewer for this comment. We corrected the number of all MS patients in Table 1 (see also A2.6) and removed the priority claims in the first paragraph of discussion: “We present a large longitudinal multicenter study analyzing changes of retinal layer thickness and visual function and their predictive power for subsequent disease activity and disability progression in RRMS, SPMS, and PPMS patients.”.

Reviewer #2

This is a very interesting study, including a large cohort of MS patients of different subtypes and HC. The main messages are the uniform decrease of thickness of both pRNFL and GCIPL in RRMS and progressive MS and the prognostic value of pRNFL, mRNFL and GCIPL thickness in regards to EDSS, relapses and MRI activity in the entire cohort. The study has several strengths, such as inclusion of patients with SPMS and PPMS, robust stats, and longitudinal assessments.

Few points to consider:

R2.1: The term "MRI activity" can be misleading as in the current study refers to both gad-enhancing lesions as well as new lesions. Please re-phrase; may use MRI progression/activity.

A2.1: We rephrased the term "MRI activity" to "MRI progression/activity" throughout the manuscript. We defined MRI progression/activity as new or enlarging T2-weighted/gadolinium-enhancing lesions.

R2.2: Please clarify if the SPMS patients have active or non-active SPMS. An EDSS score of 5.5 (1.5-8) seems somewhat low for a mean duration of 20.4 years (4-43.5). Please give more details for this group.

A2.2: Thank you for pointing this out. Both active and non-active SPMS were considered for our analysis. We added the following information to the figure legend of Table 1: “At baseline 112 patients were active based on relapses, 13 patients were active based on MRI progression/activity, and 0 patients were active based on both relapses and MRI progression/activity. 20 patients were not classifiable due to missing information regarding relapses and/or MRI progression/activity in the year before baseline OCT.”

R2.3: Were all the assessments, including VA, EDSS, VEP and MRI occurred at the same visit? Please clarify.

A2.3: Assessments of VA, EDSS, and VEP were performed at the same visit as OCT. MRI occurred only partially at the same visit as OCT. For the prediction analyses, we considered all MRIs that were performed after one OCT and before the next OCT. We added these informations to the methods and results:

“EDSS scores were always assessed by the same team of specially trained neurologists at each participating center at the same visit as OCT.“

“ Assessments of visual acuity and VEP occurred at the same visit as OCT.“

“Moreover, logistic regressions were adjusted for the time to assessment of EDSS, MRI, and relapses. This was the OCT inter-scan interval for the prediction of EDSS, the interval between OCT and each MRI for the prediction of MRI progression/activity, the interval between OCT and relapse (if relapse occurred before next OCT) or the OCT inter-scan interval (if no relapse occurred) for the prediction of relapses.”

“Information of 947 MRIs after baseline OCT, 648 MRIs after second OCT visit, 363 MRIs after third OCT visit, 178 MRIs after fourth OCT visit, 92 MRIs after fifth OCT visit, 44 MRIs after sixth OCT visit, and 2 MRIs after seventh OCT visit were included in the logistic mixed-effects regression analysis.”

R2.4: Were the patients screened for glaucoma/ophthalmological diseases by an eye specialist? Were the patients screened for hemoglobin A1C or their co-morbidities were self-reported? Please add/clarify.

A2.4: Thank you for this comment. We added the following informations to the methods: “All patients were screened for glaucoma/ophthalmological diseases by medical history. A brief ophthalmological examination for exclusion of ophthalmological diseases was performed by trained specialists namely experienced neurologists and/or optometrists. Information on co-morbidities and abnormal laboratory parameters were available based on physician's letters, diagnostic findings, and blood analyses.“

R2.5: Please explain the logic behind dividing the disease duration into certain intervals: 0-3.5, 3.6-5.5, 5.6-7.5 etc, for RRMS and PPMS; similarly the 3.5-12.5 y, etc for SPMS.

A2.5: Patients were examined at varying time points during their disease course and with different frequencies. To account for the varying influence of disease duration at the different stages of the disease, the sample was divided into subgroups according to the disease duration for longitudinal analysis. We chose these interval boundaries to achieve an equal distribution of data within the different bins, especially for the PPMS group (**Supplementary Table 5**) in which we were particularly interested. The division of the HCs and RRMS group into different subgroups was orientated to the group of PPMS. Furthermore, in response to the comments raised by reviewer 1 we added a spline fit to investigate changes over the different phases of disease in a flexible model (see **new Figure 3**).

R2.6: Table 1. The numbers do not add up for the MS patients (195+125+87=407; not 505; please revise.

A2.6: Thank you for pointing out this error. 505 was the number of all participants (healthy controls and MS patients). We corrected the number of all MS patients in Table 1 (N=407).

Reviewer #3

Kraemer et al. conducted a longitudinal study analyzing changes of retinal layer thickness and visual function over time and their predictive value for subsequent disease activity and disability progression in a relatively large cohort of RRMS, SPMS, and PPMS patients. The objective of identifying OCT measures that predict patients at high risk of disease activity/progression is extremely relevant. Results show that pRNFL and GCIPL are robust markers of neuroaxonal damage of the visual pathway throughout the disease course in both relapsing and progressive MS and that SD-OCT assessment performed at a single time point can predict future disease activity and disability progression. Some quite unique aspects for this study are: (I) the high proportion of progressive patients, (II) the fact that SPMS and PPMS were considered separately, and (III) the large number of follow-up assessments (leading to the observation that longitudinal assessments of retinal thickness are not suitable on a single patient level). Overall, the study is relevant and well conducted, although largely confirmatory of prior studies. The following issues should be addressed:

R3.1: The authors state that 2651 measurements of 195 RRMS, 87 SPMS, 125 PPMS patients, and 98 HCs were included in the analysis; however, it should be specified how many OCT/MRI were available for each timepoint and how many patients had 2, 3 or more timepoints; this information could be added to the supplementary material or specified in Fig. 1.

A3.1: Thank you for this helpful comment which is important for interpreting our results. OCT assessments were performed at varying time points and with different frequencies. We now specified the number of available OCT measurements of each visit in Figure 1 and the mean number of patients with 2, 3, and more OCT visits in the results: “505 individuals ($N_{RRMS} = 195$, $N_{SPMS} = 87$, $N_{PPMS} = 125$, $N_{HCs} = 98$) had two OCT measurements, 263 individuals ($N_{RRMS} = 124$, $N_{SPMS} = 41$, $N_{PPMS} = 60$, $N_{HCs} = 38$) three OCT measurements, 132 individuals ($N_{RRMS} = 86$, $N_{SPMS} = 13$, $N_{PPMS} = 24$, $N_{HCs} = 9$) four OCT measurements, 64 individuals ($N_{RRMS} = 49$, $N_{SPMS} = 6$, $N_{PPMS} = 8$, $N_{HCs} = 1$) five OCT measurements, 29 individuals ($N_{RRMS} = 23$, $N_{SPMS} = 3$, $N_{PPMS} = 3$, $N_{HCs} = 0$) six OCT measurements, and 13 individuals ($N_{RRMS} = 11$, $N_{SPMS} = 0$, $N_{PPMS} = 2$, $N_{HCs} = 0$) seven OCT measurements”.

The MRIs were only partially performed at the same visit as OCT. For the prediction analyses we considered all MRIs that occurred after one OCT and before the next OCT. We added the following sentence to the results: “Information of 947 MRIs after baseline OCT, 648 MRIs after second OCT visit, 363 MRIs after third OCT visit, 178 MRIs after fourth OCT visit, 92 MRIs after fifth OCT visit, 44 MRIs after sixth OCT visit, and 2 MRIs after seventh OCT visit were included in the logistic mixed-effects regression analysis.” We added the following sentence to the section *Clinical assessment*: “In a subset of participants ($N_{RRMS} =$

186, N_{SPMS} = 55, N_{PPMS} = 73) information on MRI progression/activity after OCT examinations were available.“

R3.2: Were spinal cord lesions considered when exploring disease activity or just brain? If only data regarding brain MRI activity were available, it should be listed as a limitation and discussed.

A3.2: Spinal MRIs were not regularly performed in clinical practice. However, findings about spinal MRIs were also considered if available. We complement the following sentence in the section *Clinical assessment*: “The following data were recorded: sex, date of birth, date of manifestation of patients’ first symptoms, EDSS scores, episodes of ON, DMTs, occurrence of relapses, brain and spinal MRI progression/activity (new and enlarging T2-weighted/gadolinium-enhancing lesions).”

R3.3: As reported in Table 1, percentage of patients on DMTs at baseline differed between groups (as expected, the percentage of treated patients was higher in RRMS as compared to the progressive groups of patients). Although authors state that investigations for the different therapeutics were judged beyond the scope of this study, we think that analyses should somehow take into consideration whether patients were treated or not (adding it as a covariate in the predictive models? sensitivity analyses?). Indeed, not only treatment strongly influences disease activity (which is one of the main outcomes explored in this study), but it has also been suggested that DMT initiation may prevent GCIPL thinning and may lead to reduction in INL thickness (Bsteh G, et al. Eur J Neurol. 2021, ref 7; Knier et al. Brain 2016, ref 13). If available, the duration of treatment for the MS patients on their DMT at the time of the OCT should be also reported. Moreover, it should be stated whether or not all patients remained in the same DMT during the follow-up and eventually those who did not should be analyzed differently.

A3.3: Thank you for this comment. The issue of treatment effects is very complex due to the considerable indication bias (patients with presumably more severe disease courses were more likely to receive high efficacy treatment). We, therefore, chose not to focus on DMTs in this paper. We plan to address treatment in a separate investigation. However, we agree with this reviewer that DMTs are relevant and have now added the exposure of DMT (high-efficacy DMT/low-efficacy DMT/no DMT) at OCT visit as a covariate in the predictive models. We added the following section to the methods: “Prognostic models of EDSS progression, MRI progression/activity, and relapses (**Table 2**) were calculated using binary logistic mixed effects models with eye-specific intercepts. This approach allowed us to account for the hierarchical structure of our data with two eyes per subject, repeated OCT measurements within subjects, and individual differences in baseline values. We adjusted the analyses for disease duration, age and ON (categorized in the categories a) ON based on medical history, b) ON based on inter-eye GCIPL thickness difference ⁴³, and c) no ON), sex, baseline EDSS, DMT (high-efficacy DMT, low-efficacy DMT, and no DMT) if necessary, as indicated by a significant likelihood-ratio-test in favor of the more complex model.“

R3.4: One relevant finding of this study is that pRNFL, macular RNFL, and GCIPL thickness predicted future disease activity. However, this correlation was evident when patients with RRMS, SPMS, and PPMS were included in the analysis all together, while analyzing the MS

subtypes separately revealed no significant prediction. According to the authors' discussion, it could be due to insufficient sample size for the subgroups. On this matter, due to the retrospective nature of the study including patients from 5 different German centers, authors decided to group MS individuals into subgroups with variable disease duration intervals to achieve an equal distribution of data among cohorts introducing, in my view an arbitrary bias in the analysis. It should be considered however that other papers with a similar sample size found an association between OCT metrics and disease activity/progression (Bsteh G, et al. Eur J Neurol. 2021, ref 7; Cellierino et al, J Neuroophthalmol. 2021).

A3.4: Thank you for this comment. The division of patients into subgroups according to the disease duration was not used in the predictive analyses of EDSS progression, MRI progression/activity, and relapses. We completely revised our prediction analyses by adjusting for important covariates as ON (categorized in the categories a) ON based on medical history, b) ON based on inter-eye GCIPL thickness difference, and c) no ON) (reviewer comment R1.5), baseline EDSS (reviewer comment R3.5), DMT (high-efficacy DMT, low-efficacy DMT, and no DMT) (R3.3), and time to assessment of EDSS, MRI, and relapses (R1.8). Moreover, we performed Kaplan-Meier analysis with Cox proportional hazards models to assess the value of each OCT parameter at baseline as a potential risk factor for suffering disability worsening, MRI progression/activity, and relapses."

We added the following sentences to the methods: "Prognostic models of EDSS progression, MRI progression/activity, and relapses (**Table 2**) were calculated using binary logistic mixed effects models with eye-specific intercepts. This approach allowed us to account for the hierarchical structure of our data with two eyes per subject, repeated OCT measurements within subjects, and individual differences in baseline values. We adjusted the analyses for disease duration, age and ON (categorized in the categories a) ON based on medical history, b) ON based on inter-eye GCIPL thickness difference ⁴³, and c) no ON), sex, baseline EDSS, DMT (high-efficacy DMT, low-efficacy DMT, and no DMT) if necessary, as indicated by a significant likelihood-ratio-test in favor of the more complex model. Moreover, logistic regressions were adjusted for the time to assessment of EDSS, MRI, and relapses. This was the OCT inter-scan interval for the prediction of EDSS, the interval between OCT and each MRI for the prediction of MRI progression/activity, the interval between OCT and relapse (if relapse occurred before next OCT) or the OCT inter-scan interval (if no relapse occurred) for the prediction of relapses." "We performed Kaplan-Meier analysis with Cox proportional hazards models to assess the value of each OCT parameter at baseline as a potential risk factor for suffering disability worsening, MRI progression/activity, and relapses. The mean of the values of pRNFL, mRNFL, GCIPL, INL thickness for both eyes were used in the Kaplan-Meier analysis (**Figure 5**) and Cox regression models (**Supplementary Table 6**). For Kaplan-Meier and Cox regression analyses, we grouped patients into tertiles according to the thickness of pRNFL, mRNFL, GCIPL, or INL at the time of study enrolment. We compared the risks associated with being in the lowest tertile (pRNFL \leq 88 μ m, mRNFL \leq 29 μ m; GCIPL \leq 63 μ m; INL \leq 33 μ m) versus the two upper tertiles. The same covariates and the same stepwise feature selection as in the prediction analyses were used in the multiple Cox proportional hazard regression model. We report the hazard ratio (HR) with 95% CIs (**Supplementary Table 6**)."

R3.5: In the Cellierino manuscript the association between OCT metrics and disability worsening was statistically significant in RRMS but not in PMS patients. Similarly, Martinez-Lapiscina et al suggested that a single pRNFL assessment could be predictive of disability

worsening in MS, but they did not detect significant differences in baseline OCT features between patients with PMS whose disability worsened and those who remained stable (GCIPL data were not available at the time). In the aforementioned studies, the weak correlation between baseline OCT metrics and subsequent disability in patients was explained, at least partly, by the disproportioned retinal injury resulting from higher age, disease duration and baseline disability, which characterize PMS patients. In this paper, Kraemer et al deeply explore the association with disease duration, but never take into account baseline disability (which – according to Table 1 – ranges from 0 to 7 in the RRMS group). The possible influence of baseline disability in predicting clinical outcome over time should be explored (maybe excluding from the RRMS group patients with EDSS>5.5, scores which are atypical for a relapsing-remitting disease course). Altogether, these results should be further discussed in the context of previous findings.

A3.5: We are very grateful that we have been advised to adjust our predictive analyses for important covariates as ON (categorized in the categories a) ON based on medical history, b) ON based on inter-eye GCIPL thickness difference, and c) no ON) (reviewer comment R1.5), baseline EDSS (reviewer comment R3.5), DMT (high-efficacy DMT, low-efficacy DMT, and no DMT) (R3.3), and time to assessment of EDSS, MRI, and relapses (R1.8). By adjusting our predictive analyses for these covariates, several previous results lost significance. However, pRNFL and mRNFL still predicted future relapses in all MS and RRMS patients while mRNFL, GCIPL, and INL thickness predicted future MRI activity in PPMS. We added the following section to the methods, results and discussion: “We adjusted the analyses for disease duration, age and ON (categorized in the categories a) ON based on medical history, b) ON based on inter-eye GCIPL thickness difference ⁴³, and c) no ON), sex, baseline EDSS, DMT (high-efficacy DMT, low-efficacy DMT, and no DMT) if necessary, as indicated by a significant likelihood-ratio-test in favor of the more complex model. Moreover, logistic regressions were adjusted for the time to assessment of EDSS, MRI, and relapses. This was the OCT inter-scan interval for the prediction of EDSS, the interval between OCT and each MRI for the prediction of MRI progression/activity, the interval between OCT and relapse (if relapse occurred before next OCT) or the OCT inter-scan interval (if no relapse occurred) for the prediction of relapses.”

“When adjusting for all covariates in the logistic mixed-effects regression, lower pRNFL and mRNFL thickness were associated with increased probability for relapses in all MS patients and those without ON (pRNFL and mRNFL), and in all RRMS patients and those without ON (pRNFL) (**Table 2**).

mRNFL, GCIPL and INL thickness predicted future MRI progression/activity in PPMS patients. , Thicker INL and paradoxically thicker mRNFL were associated with increased probability for MRI progression/activity in SPMS. The significance levels and odds ratios are provided in **Table 2**. As an example, an odds ratio of 0.62 for the pRNFL or mRNFL regarding relapses in MS patients, respectively means that each 1 μm of pRNFL or mRNFL atrophy is associated with an increase of the odds for disability progression by 38% for relapses in MS patients (**Table 2**).”

“In line with a previous study ¹², pRNFL and mRNFL were predictive of relapses in MS and RRMS. In contrast to previous studies ^{4-6,9,11,15,38}, OCT measurements were not associated with future disability progression in our study (**Table 2**). Reasons for this could be the mean short time to assessment of EDSS of 1.83y ($\pm 1.53\text{y}$) and the fact that we included a heterogeneous group of patients with different disease duration at baseline OCT (**Table 1**) and different intervals of assessments in contrast to previous studies, which focused on patients with predominantly early MS. While several previous studies analyzed associations

between baseline OCT and subsequent disability progression and/or disease activity^{4,5,16,17}, we also included longitudinal follow-up assessments adjusting for disease duration and time to assessment (**Table 2**).

Interestingly, lower mRNFL, GCIPL and INL thickness were only associated with increased risk for MRI progression/activity in PPMS patients (**Table 2**) suggesting that retinal atrophy in PPMS is driven by inflammation (new and enlarging T2-weighted/gadolinium-enhancing lesions) and has predictive value. The association of higher mRNFL with increased probability for MRI progression/activity in SPMS is an unexpected and counterintuitive finding which is perhaps caused by the heterogeneity of our cohort and may be the result of a statistical artefact. The association of higher INL thickness with increased risk for MRI progression/activity in SPMS might be suggestive of inflammatory processes in this layer, which have been observed in RRMS¹³, and might also play a role in SPMS.“

Only 2 of 195 RRMS patients (1.14%) had an EDSS>5.5 at baseline OCT. When excluding OCT visits of those patients, the results of our predictive analyses did not change.

R3.6: When authors explore the association with disability worsening, it should be specified whether EDSS progression was assessed at a single time-point or confirmed (i.e., EDSS assessment repeated at 3 or 6 months to confirm if disability progression was “sustained”).

A3.6: Thank you for this comment, which was also raised by reviewer 1. We added the following sentence to the methods: “Disability worsening was defined as a documented increase in EDSS score as a consequence of relapses or disease progression (≥ 1.0 point in case baseline EDSS score was < 6.0 , or ≥ 0.5 point if baseline EDSS score was ≥ 6.0) at a single time point“. The term “confirmed EDSS worsening” in the results section was incorrect and misleading. Therefore, we deleted the word “confirmed” before EDSS worsening (see also A1.9).

R3.7: Were there additional exclusion criteria that were not mentioned in this manuscript? Prior/concomitant administration of certain chemotherapeutic agents should be considered as exclusionary criteria (as it actually seems to be the case when looking at Fig. 1); however, iatrogenic optic neuropathy is not mentioned in the text.

A3.7: Thank you very much for pointing this out. Treatment with substances with increased risk of iatrogenic retinopathy was, in deed, an additional exclusion criterion, which we failed to mention. We added this information to the methods: “Exclusion criteria were any diseases of the optic nerve or retina not related to MS; a diagnosis of other neuroinflammatory disorders (i.e., neuromyelitis optica spectrum disorders); severe refraction anomalies $\geq \pm 6$ diopters; systemic conditions that could affect the visual system; treatment with substances with increased risk of iatrogenic retinopathy such as chemotherapy; insufficient scan quality according to the OSCAR-IB criteria^{40,41} (**Figure 1**). In MS patients, initial swelling and retinal atrophy in the context of acute ON has a major impact on retinal layer thickness⁴². For this reason, we excluded eyes with previous ON within 6 months to baseline OCT and those with ON between OCT measurements (**Figure 1**).“

R3.8: In the methods section, the authors define the study “exploratory”; accordingly, they did not adjust for multiple comparison. Since several papers have already explored changes in

terms of OCT metrics over time and their predictive value in MS (even if with a lower percentage of progressive patients), authors should avoid defining this study “exploratory”.

A3.8: We thank the reviewer for this comment. We deleted the word “exploratory”.

References

1. Hamdy E, et al. Diagnosing ‘transition’ to secondary progressive multiple sclerosis (SPMS): A step-by-step approach for clinicians. *Mult Scler Relat Disord* 60; 103718 (2022).
2. Baecher-Allan C, et al. Multiple Sclerosis: Mechanisms and Immunotherapy. *Neuron* 97(4); 742-768 (2018).
3. Balk, L.J. et al. Timing of retinal neuronal and axonal loss in MS: a longitudinal OCT study. *Journal of neurology* 263, 1323-1331 (2016).

Reviewers' Comments:

Reviewer #1 (Remarks to the Author):

I thank the authors for their response and their sincere attempt to address points raised in my prior review. Unfortunately, there remain significant issues and I have concerns that the statistical approach utilized is inappropriate and renders the authors' interpretation of the coefficients invalid, and thus requires a complete re-analysis and re-interpretation of the data. While there are often many different ways to approach a specific analysis and there is a subjective component to the approach chosen, the current approach objectively contains obvious and critical flaws, as I outline below.

In the authors' response it has become clear that the authors are including both time-varying age and follow-up time in the models. This is clearly inappropriate and the interpretation of the coefficients by the authors is incorrect in this context. The coefficient for age in this setting captures both cross-sectional and longitudinal effects of age (as the variable is time-varying) on the dependent variable, whereas follow-up time also captures longitudinal effects (as it is also increasing over time). Thus, it is impossible to interpret the coefficient for follow-up time as representing the annualized change in the dependent variable, as this information is also captured by the coefficient for the age variable (together with cross-sectional effects of age). The authors should seek input from a statistician with expertise in longitudinal analysis. Please refer to PMID 19196902, which can help provide the authors with further insight regarding the problems with their statistical approach.

Furthermore, consistent with my prior recommendation, the authors have now attempted to model variables such as disease duration using non-linear approaches. However, only very limited information is provided about the spline fit, with unclear spline basis, knot placement (and selection), and whether natural splines were used. Moreover, in the longitudinal analyses incorporating splines, it seems that these were fitted incorrectly and notably the figures do not demonstrate clearly consistent findings with the binned analysis, nor are they presented in a similar fashion. For example, the authors report that the pRNFL is decreasing over time in the controls (median rate of change -0.13), but in Figure 3A it appears that the pRNFL thickness is increasing over time in controls based on the spline fit that is overlaid. Thus, it seems that a regression was simply fit to all data points for the figure without accounting for the correlated nature of the data (given repeated measures for individual eyes). The spline variables would need to be fit in a mixed effects model and the estimated regression line would then need to be plotted. Furthermore, the panels in Figure 3 showing the spaghetti plots with overlaid fits have a y-axis that is the actual measure (e.g. pRNFL), whereas the boxplots for the binned analysis have a y-axis that is the annualized rate of change in the measure (which is the derivative of the lines showed in the spaghetti plot panels), so it is not clear that the findings are actually consistent between the two approaches. The spline regression need to be fitted correctly (with a mixed effects model) and presented appropriately so that the reader can compare with the findings of the binned analysis.

Reviewer #2 (Remarks to the Author):

All my comments were successfully addressed. Thank you for revising your manuscript.

Reviewer #3 (Remarks to the Author):

The manuscript has been fully revised addressing properly not only my comments but also those from the other reviewers. This makes the results presented in this paper much more consistent and relevant. I do not have any other relevant comment and I support the publication of the manuscript.

Reviewer #1

R1.1: I thank the authors for their response and their sincere attempt to address points raised in my prior review. Unfortunately, there remain significant issues and I have concerns that the statistical approach utilized is inappropriate and renders the authors' interpretation of the coefficients invalid, and thus requires a complete re-analysis and re-interpretation of the data. While there are often many different ways to approach a specific analysis and there is a subjective component to the approach chosen, the current approach objectively contains obvious and critical flaws, as I outline below.

In the authors' response it has become clear that the authors are including both time-varying age and follow-up time in the models. This is clearly inappropriate and the interpretation of the coefficients by the authors is incorrect in this context. The coefficient for age in this setting captures both cross-sectional and longitudinal effects of age (as the variable is time-varying) on the dependent variable, whereas follow-up time also captures longitudinal effects (as it is also increasing over time). Thus, it is impossible to interpret the coefficient for follow-up time as representing the annualized change in the dependent variable, as this information is also captured by the coefficient for the age variable (together with cross-sectional effects of age). The authors should seek input from a statistician with expertise in longitudinal analysis. Please refer to PMID 19196902, which can help provide the authors with further insight regarding the problems with their statistical approach.

A1.1: We apologize for not making this point clear enough. When reading our point-by-point response for the second time, we realized that our answer may have been misleading. We included time-varying age but not follow-up time in the models. Instead we adjusted our predictive analyses for the time from OCT to assessment of EDSS, MRI, and relapses to account for your previous comments (see R1.8. in previous response letter). Both covariates time-varying age and time to assessment do not relate to each other because the time to assessment, meaning time from each OCT to each measurement, does not depend on age.

For clarification, we have specified this in the methods: "Prognostic models of EDSS progression, MRI progression/activity, and relapses (**Table 2**) were calculated using binary logistic mixed effects models with eye-specific random intercepts. This approach allowed us to account for the hierarchical structure of our data with two eyes per subject, repeated OCT measurements within subjects, and individual differences in baseline values. We adjusted the analyses for disease duration or participants' age at assessment (depending on fit), ON (categorized in the categories a) ON based on medical history, b) ON based on inter-eye GCIPL thickness difference ⁴³, and c) no ON), sex, baseline EDSS, DMT (high-efficacy DMT, low-efficacy DMT, and no DMT) if necessary, as

indicated by a significant likelihood-ratio-test in favor of the more complex model. Moreover, logistic regressions were adjusted for the time to assessment of EDSS, MRI, and relapses. This was the OCT inter-scan interval for the prediction of EDSS, the interval between OCT and each MRI for the prediction of MRI progression/activity, the interval between OCT and relapse (if relapse occurred before next OCT) or the OCT inter-scan interval (if no relapse occurred) for the prediction of relapses.” Moreover, logistic regressions were not adjusted for follow-up time, but for the time to assessment of EDSS, MRI, and relapses. All analyses were performed by two statisticians with expertise in longitudinal analysis (CB and AH) who are coauthors of the manuscript.

R1.2: Furthermore, consistent with my prior recommendation, the authors have now attempted to model variables such as disease duration using non-linear approaches. However, only very limited information is provided about the spline fit, with unclear spline basis, knot placement (and selection), and whether natural splines were used.

A1.2: We thank the reviewer for this important comment. We now explained the applied method for creating splines in the section *Statistical analysis* of the manuscript: “As an alternative statistical approach to model longitudinal effects of disease duration (and age) on retinal layers thickness, VEP latency, and VA, b-splines were fitted in a mixed model with random intercept for eyes and controlling for gender. Knots were placed in equally spaced percentiles (e.g. knots were placed at the 25%, 50% and 75% percentiles when using three knots). The number of knots and polynomial degrees (both between 1 and 5) were chosen such that 10-fold cross validation mean squared error (MSE) was minimized. This was done for each disease course and dependent variable separately.” We described the applied method also in the legend of **Figure 3** “B-splines were fitted in a mixed model with random intercept for eyes and controlling for gender. The number of knots and polynomial degrees (both between 1 and 5) were chosen such that 10-fold cross validation mean squared error (MSE) was minimized.” and in the legend of **Supplementary Figure 3** “B-splines were fitted in a mixed model with random intercept for eyes and controlling for gender. The number of knots and polynomial degrees (both between 1 and 5) were chosen such that 10-fold cross validation mean squared error (MSE) was minimized.”

R1.3: Moreover, in the longitudinal analyses incorporating splines, it seems that these were fitted incorrectly and notably the figures do not demonstrate clearly consistent findings with the binned analysis, nor are they presented in a similar

fashion. For example, the authors report that the pRNFL is decreasing over time in the controls (median rate of change -0.13), but in Figure 3A it appears that the pRNFL thickness is increasing over time in controls based on the spline fit that is overlaid. Thus, it seems that a regression was simply fit to all data points for the figure without accounting for the correlated nature of the data (given repeated measures for individual eyes). The spline variables would need to be fit in a mixed effects model and the estimated regression line would then need to be plotted.

Furthermore, the panels in Figure 3 showing the spaghetti plots with overlaid fits have a y-axis that is the actual measure (e.g. pRNFL), whereas the boxplots for the binned analysis have a y-axis that is the annualized rate of change in the measure (which is the derivative of the lines showed in the spaghetti plot panels), so it is not clear that the findings are actually consistent between the two approaches. The spline regression need to be fitted correctly (with a mixed effects model) and presented appropriately so that the reader can compare with the findings of the binned analysis.

A1.3: We agree that the linear regression analysis with binned intervals are a completely different approach to the polynomial b-splines models albeit they analyzed the same data and addressed related questions. We showed both models because they are complementary and adding to the analyses and understanding of the data.

The previous splines disregarded the fact that the dataset included repeated measurements within subjects and therefore may not have represented the true underlying relationship between time and the dependent variables. We have now refitted the b-splines using a mixed model with random intercept for eyes and controlling for gender. The findings are now more in line with the findings of the linear regression analysis (boxplots of new Figure 3). We corrected the figures accordingly (new Figure 3, new Supplementary Figure 3). To clarify the meaning of the binned analyses, we added informations on how the boxplots were generated in the section *Statistical analysis* of the manuscript "For each disease type, we calculated the annualized thickness change rate of the different layers for the different intervals of disease duration as the difference between two consecutive thickness measurements, divided by the time between the measurements in years. In order to obtain parameter estimates for the annualized thickness change rate for the different intervals of disease duration (colored boxplots in **Figure 3A-D, Supplementary Table 4**) and for the overall annualized thickness change rate over time (boxplots bordered in red **Figure 3A-D, Supplementary Table 3**), retinal layer thickness was predicted based on the main effect of disease duration with a random intercept per eye nested in subjects and a random slope per subject. If this complex model resulted in a singular fit or nonconvergence of the model, it was reduced to a simpler model

eliminating the least relevant part(s) of the overly complex model based on the smallest variance of the random effect. Subsequently, participants' age at assessment and sex were added as covariates and likelihood-ratio-tests were carried out to select the best model (**Supplementary Table 3 and 4**)." and in the legend of **Figure 3** "For each time of assessment, the annualized thickness change rate since the last assessment was calculated based on the following formula: annualized thickness change rate= (retinal layer thickness_{current assessment} - retinal layer thickness_{last assessment}) / time since last assessment in years).".

Reviewer #2

R2.1: All my comments were successfully addressed. Thank you for revising your manuscript.

A2.1: We thank Reviewer #2 for this positive feedback on our manuscript.

Reviewer #3

R3.1: The manuscript has been fully revised addressing properly not only my comments but also those from the other reviewers. This makes the results presented in this paper much more consistent and relevant. I do not have any other relevant comment and I support the publication of the manuscript.

A3.1: We thank Reviewer #3 for this positive feedback on our manuscript.

REVIEWER COMMENTS

Reviewer #1 (Remarks to the Author):

Despite having read now three iterations of the same manuscript, it remains unclear to me how the majority of the statistical analyses were performed. The description of the methods continues to be unclear and insufficient. Given the above, I recommend that the authors actually write out the models/equations that were used for each analysis and publish the analysis code to accompany the manuscript.

Despite the above issues with the presentation of the methods, one aspect of the analysis has become somewhat clearer (perhaps). It seems now that rates of change were generated by subtracting a baseline from a follow-up retinal measurement and dividing by the duration of follow-up (notably this simple point was not clear in the two prior iterations of the manuscript). However, this approach has significant drawbacks. OCT measurements exhibit significant random variability, which exceeds the average amount of longitudinal change observed. This problem, paired with the relatively small number of observations in each stratum of disease duration leads to marked variability/noise in the longitudinal change estimates (as can easily be seen by inspecting the presented boxplots of rate of change shown in Figure 3). Furthermore, it is unclear how thickness change rates were calculated for those with more than 2 visits. If intermediate timepoints were discarded, this would decrease the precision of the estimates of the rate of change. Alternatively, one could fit an ordinary least squares model to the observations from each individual eye, which will lead to utilization of the intermediate time-points, but the variability will still be expected to be relatively high, especially in individuals with a small number of OCT evaluations. In this context, fitting a mixed effects model to the population (with the dependent variable being the retinal layer thickness measurements and follow-up time as the independent variable, with a random intercept per eye nested in subjects and a random slope) and using the empirical Bayes estimates of best linear unbiased predictions (BLUPs) would be an approach with less sensitivity to outliers/random variability (see for example PMID: 21423039), and I suggest that this also be performed as a sensitivity analysis to assess the robustness of the findings. Notably, it seems that this may have been done, but this is described as including disease duration in the model and immediately following a discussion in the methods section outlining that thickness measurements were subtracted and divided by follow-up time.

Reviewer #4 (Remarks to the Author):

Comments:

- 1) General: In many places, you report an estimate, confidence interval and p-value, but there are places with only the estimate and p-value. Please add confidence intervals in all places.
- 2) Page 6/Line 121 and Figure 2: In this analysis, it appears that you are fitting separate models for each of the groups and then stating which of the slopes are statistically significant. Could you compare the

slopes between the groups using interaction terms because a difference in significance is not the same as a difference? As an example, the estimated slope in 2b was very similar for SPMS and PPMS groups, but one is significant, and the other is not. I think the conclusion here is that the association is similar rather than different between these groups.

3) Supplementary Table 2: As with the previous comment, could a model including an interaction term be used here to estimate the group specific values as well as compare the groups? I am also uncertain why the regression coefficient for pRNFL in PPMS would decrease compared to Figure 2, but the p-value would get much smaller.

4) Page 7/Line 153: You state that significant loss of pRNFL occurred through the disease course in RRMS and PPMS, but the change was positive in the 5.6-7.5 year interval for RRMS. Is this sign correct in Supplementary Table 4?

5) Page 7/Line 157: As with my previous comments, you comment almost exclusively on the intervals with a p-value less than 0.05, but some of the intervals have only minor differences in the estimates despite large differences in the p-values. As an example, you state that mRNFL atrophy was observed in RRMS <3.5 years, but the estimated change is much smaller than in the 7.6-10.5 interval. It seems that a comparison between the groups/intervals or with the HCs would be more informative. It seems that models that included interaction terms would allow direct comparisons of the groups and intervals. Further, the comparison across the time intervals using a piecewise linear spline model would indicate whether the linear model described in Supplementary Table 3 is an appropriate fit.

6) Page 9/Line 199: I am confused by the interpretation of the OR. You state that 1 microgram of atrophy is associated with an increase in the odds of disability progression by 38%. I think the OR would mean that a one unit increase leads to a 38% reduction in the odds but inverting the OR would be an OR of 1.62. Also, is this the OR for a relapse occurring or disability progression due to relapses?

7) Page 9: I do not understand why the results would be so different comparing the mixed effects logistic regression model and the Cox proportional hazards model for relapse with pRNFL (Table 2 vs. Supplementary Table 6). Could you explain the difference in the modeling assumptions since both approaches are analyzing the impact of pRNFL on the likelihood of a relapse?

8) Page 9/Line 214 and Line 221: Were the inconclusive results not significant or were the confidence intervals too wide to be able to make a statement? I believe this should be clarified. If the results were just not significant, I think this is different than inconclusive.

9) Page 15/Line 405 and Table 1: Was a chi-squared test used to compare age between groups?

10) Table 1: The listed IQR for the disease duration for RRMS patients seems to have an error since the IQR is from 0-30.

11) Page 16: Reviewer 1 commented on the potential importance of distinguishing between cross-sectional and longitudinal effects, but the proposed model does not distinguish between these. Did you test whether the longitudinal and cross-sectional effects of disease duration are equal as requested by Reviewer 1?

12) Page 16: Is your model including disease duration at assessment, which is changing with time, and age at assessment, which is also changing with time? This would seem to include two measures of within subject change. Also, it is also not clear which of the results in Supplementary Tables 3 and 4 included age and gender. Please clarify in the table legends which model was fit for each model.

13) Page 16: I understand that the random effects were chosen based on likelihood ratio tests, but I am a little confused by the choice of the random effects. It seems that there is a random intercept for eye and no random slope for eye, and a random slope for subject but no random intercept for subject. Is that

correct?

14) Page 16: When subjects had measurements that spanned multiple intervals, how was this handled?

15) Page 17, Line 452: Did you use a Gaussian link or an identity link?

16) Page 17: For the analysis of EDSS progression, was the progression indicator (Y/N) a comparison of the EDSS to the previous EDSS measurement or the baseline EDSS measurement? Was the EDSS progression required to be sustained?

17) Page 17: For the Cox model, was only time to first relapse analyzed or was it the time to each subsequent relapse? If there were multiple events, how were the recurrent events accounted for?

Reviewer #1:

R1.1: Despite having read now three iterations of the same manuscript, it remains unclear to me how the majority of the statistical analyses were performed. The description of the methods continues to be unclear and insufficient. Given the above, I recommend that the authors actually write out the models/equations that were used for each analysis and publish the analysis code to accompany the manuscript.

A1.1: We thank you for this comment. We thoroughly revised the manuscript and now describe the statistical analyses in more detail. Each time results are presented, we now indicate which statistical tests were employed. For a better understanding of our analyses, we have deposited the statistical codes. We apologize that we have not been clear enough in describing our statistical approaches in the previous versions of the manuscript. We understand that, in the previous versions, it was difficult for readers to match the different statistical models described in our methods section to the results, figures and tables.

We have involved Dr. Joachim Gerß, PHD, as third statistician. As deputy director of the Institute for Biometrics and Clinical Research of the University Muenster he has outstanding expertise in longitudinal analysis. He checked all analyses and played a consultative role in the calculation of empirical Bayes estimates of best linear unbiased predictions (BLUPs).

We added the following sentences to the methods to make the statistical approaches clearer:

“All statistical analyses were performed using the best fit LMM identified by likelihood ratio tests using restricted maximum likelihood approach (SPSS Statistics 26.0 (IBM) or the lme4 package in R Studio (version 1.3.1093)).”

“First, we investigated the effect of disease duration on retinal layer thickness, VEP latency, and VA at baseline by predicting the former based on a fixed effect for disease duration and random slope for person (**Figure 2**). We calculated a separate model for each of the groups. To account for confounding influences, age and/or sex (**Figure 2**) and age, sex, and/or ON (categorized in the categories a) ON based on medical history, b) ON based on inter-eye GCIPL thickness difference⁴³, and c) no ON) (**Supplementary Table 2**) were subsequently added to the model if the model fit could be improved. All models were compared based on likelihood-ratio-tests.

An interaction term with disease duration and disease course as well as participants' age, sex, and/or history of ON was included and its contrast-effects were checked using an F-test. If that test revealed significant differences, estimated marginal means of linear trends were used to test differences in atrophy rates between disease courses for significance (using Tukey's method).“

“In **Figure 3A-D**, we calculated the annualized thickness change rates for the different intervals of disease duration (colored boxplots) for each disease type as the difference between two consecutive thickness measurements, divided by the time between the measurements in years. If more than two OCTs were in one interval, several annualized thickness change rates were entered for this subject in the interval. In cases with measurement intervals spanning several disease duration intervals the calculated annualized thickness change rate was considered for the interval of the follow up scan.

Further, we calculated the overall annualized thickness change rates of the different layers in μm (boxplots bordered in red) by calculating the average of all annualized thickness change rates of the different intervals of disease.

These binned analyses were only used to obtain the boxplots displaying medians and interquartile ranges in **Figure 3A-D** for graphical representation. LMM analyses were used for hypothesis testing (calculation of the significances and change rates presented in **Supplementary Tables 3 and 4**). In order to obtain parameter estimates for the annualized thickness change rate for the different intervals of disease duration (**Supplementary Table 4**) and for the overall annualized thickness change rate over time (**Supplementary Table 3**), retinal layer thickness was predicted based on the main effect of disease duration with a random intercept per eye nested in subjects and a random slope per subject using a LMM. A random slope was only introduced for subject, as, naturally, disease duration did not vary between eyes but only between subjects. Therefore, no random slope for eye was included in the model. The annualized thickness change rate estimates were derived from the parameter estimate of the main effect of disease duration.

If this complex model resulted in a singular fit or nonconvergence of the model, it was reduced to a simpler model eliminating the least relevant part(s) of the overly complex model based on the smallest variance of the random effect. Subsequently, participants' age at assessment and sex were added as covariates and likelihood-ratio-tests were carried out to select the best model (**Supplementary Table 3 and 4**).

In order to account for the high variability of the thickness measurements, we performed empirical Bayes estimates of BLUPs of the longitudinal effect of disease duration (subtracting the baseline disease duration for each patient from the disease duration at assessment) with time since baseline, age and sex as predictors, reporting histograms of estimates across disease courses (**Supplementary Table 6**) and eye wise OLS with disease duration as only predictor and aggregated estimated regression coefficients (**Supplementary Table 7**). To decrease the high variability associated with the small number of observations in each stratum of disease duration, coefficients were estimated over the entire disease duration. To test differences between the disease courses, we ran an ANOVA on empirical Bayes estimates of BLUPs of annualized thickness change rates (**Supplementary Table 6**), using post hoc tests (Tukey) to determine which groups differed with respect to their atrophy rates.

To further reduce the variability of OLS estimates, we discarded eyes with less than three OCTs (follow-up median (IQR): 3.95 years (2.97 – 6.03 years), **Supplementary Figure 5**). To test the aggregated coefficients for significance, Wilcoxon rank-sum tests were run for each outcome and disease course. To test whether there are significant differences between disease courses, we performed an ANOVA with random intercepts per subject. In cases where the ANOVA revealed significant group differences, we conducted a Tukey post-hoc test to identify which groups differed significantly.

While overall fixed effect estimates (**Supplementary Tables 3 and 4**) capture both cross-sectional and longitudinal effects, described BLUPs (**Supplementary Table 6**) and eye wise OLS (**Supplementary**

Table 7) capture exclusively longitudinal effects of disease duration and therefore provide an additional perspective.”

“The following analyses were corrected for participants’ age and/or sex if the model fit could be improved. This is indicated in all figures and/or tables presenting results.”

“All continuous covariates were standardized in order to yield a more balanced optimization problem for model fitting.”

“For the Cox model, only the time to first relapse, MRI activity or disability worsening (see methods) after baseline was analyzed. Recurrent events were not considered.”

Moreover, we have added the following section to the discussion to explain the differences and advantages of the different applied models to analyze the effect of time (disease duration) on retinal layer thickness: “In order to analyze the effect of time (disease duration) on retinal layer thickness, we present multiple approaches, each providing a unique perspective on estimation of retinal thickness change rates. With the baseline LMMs (**Figure 2**), we present an exclusively cross-sectional approach to estimate thickness change rates and complement it using non-linear spline regressions to capture potential non-linear thickness change rates (**Supplementary Figure 1**). Furthermore, we present two approaches that measure only longitudinal effects, with the BLUPs being based on an overall LMM (**Supplementary Figure 4, Supplementary Table 6**) and the eye-wise OLS on separate models for each eye (**Supplementary Figure 5, Supplementary Table 7**). Thus, BLUPs of thickness change rates consider a larger sample by also respecting overall thickness change rates, while the OLS approach focuses exclusively on estimating overall thickness change rates for each eye separately. Therefore, BLUPs provide a lower variance estimation of thickness change, which also causes smaller differences in estimates across patients. These differences in thickness change are better captured by eye-wise OLS, which in turn suffers from higher variance in estimates.

In addition to providing both complementary views on the estimation of thickness change rates (longitudinal and cross-sectional), we also present models that include both types of effects. Namely, we present a regular LMM (**Supplementary Table 3**) estimating linear thickness change rates across the entire sample as well as a LMM with the non-linear spline representation of the effect of disease duration (**Figure 3**). Therefore, our study provides an exhaustive analysis of retinal thickness change rates across MS patients of different subgroups.”

R1.2: Despite the above issues with the presentation of the methods, one aspect of the analysis has become somewhat clearer (perhaps). It seems now that rates of change were generated by subtracting a baseline from a follow-up retinal measurement and dividing by the duration of follow-up (notably this simple point was not clear in the two prior iterations of the manuscript). However, this approach has significant drawbacks. OCT measurements exhibit significant random variability, which exceeds the average amount of longitudinal change observed. This problem, paired with the relatively small number of observations in each stratum of disease duration leads to marked variability/noise in the longitudinal change estimates (as can easily be seen by inspecting the presented boxplots of rate

of change shown in Figure 3). Furthermore, it is unclear how thickness change rates were calculated for those with more than 2 visits. If intermediate timepoints were discarded, this would decrease the precision of the estimates of the rate of change. Alternatively, one could fit an ordinary least squares model to the observations from each individual eye, which will lead to utilization of the intermediate time-points, but the variability will still be expected to be relatively high, especially in individuals with a small number of OCT evaluations.

A1.2: We thank the reviewer for this helpful comment. We agree that the binned analyses calculating the thickness change rate between two OCTs have significant drawbacks such as high variability. However, this analysis was only used to obtain the boxplots displaying medians and interquartile ranges in **Figure 3A-D** for descriptive purposes while linear mixed-effects models (LMM) were used for hypothesis testing (calculation of the significances and change rates presented in **Supplementary Tables 3 and 4**).

Furthermore, in response to this reviewer's concerns and in order to account for the high variability of the thickness measurements, we performed empirical Bayes estimates of BLUPs of the longitudinal effect of disease duration with time since baseline, age and sex as predictors and eye wise OLS with disease duration as only predictor and aggregated estimated regression coefficients. To further reduce the variability of OLS estimates, we discarded eyes with less than three OCTs.

We added the following section to the methods of the manuscript and submitted new **Supplementary Figures 4 and 5** and new **Supplementary Tables 6 and 7**: "In order to account for the high variability of the thickness measurements, we performed empirical Bayes estimates of BLUPs of the longitudinal effect of disease duration (subtracting the baseline disease duration for each patient from the disease duration at assessment) with time since baseline, age and sex as predictors, reporting histograms of estimates across disease courses (**Supplementary Table 6**) and eye wise OLS with disease duration as only predictor and aggregated estimated regression coefficients (**Supplementary Table 7**). To decrease the high variability associated with the small number of observations in each stratum of disease duration, coefficients were estimated over the entire disease duration. To test differences between the disease courses, we ran an ANOVA on empirical Bayes estimates of BLUPs of annualized thickness change rates (**Supplementary Table 6**), using post hoc tests (Tukey) to determine which groups differed with respect to their atrophy rates.

To further reduce the variability of OLS estimates, we discarded eyes with less than three OCTs (follow-up median (IQR): 3.95 years (2.97 – 6.03 years), **Supplementary Figure 5**). To test the aggregated coefficients for significance, Wilcoxon rank-sum tests were run for each outcome and disease course. To test whether there are significant differences between disease courses, we performed an ANOVA with random intercepts per subject. In cases where the ANOVA revealed significant group differences, we conducted a Tukey post-hoc test to identify which groups differed significantly.

While overall fixed effect estimates (**Supplementary Tables 3 and 4**) capture both cross-sectional and longitudinal effects, described BLUPs (**Supplementary Table 6**) and eye wise OLS (**Supplementary**

Table 7) capture exclusively longitudinal effects of disease duration and therefore provide an additional perspective.”

Since the OLS analysis has significant drawbacks as it cannot adjust for covariates that are constant within eyes, we presented the results only as **Supplementary Figure 5** and **Supplementary Table 7** and added the following section to the results: “Wilcoxon rank-sum tests of eye wise ordinary least squares (OLS) coefficients for disease duration with retinal layer thickness as outcome variable demonstrated significant atrophy of all retinal layers of all groups, except for SPMS with regard to pRNFL, mRNFL, and INL and PPMS with regard to INL (**Supplementary Figure 5**). ANOVA of eye wise OLS atrophy rate estimations with random intercept per subject showed significant differences of atrophy rates across disease courses for pRNFL ($F(2,186)=4.14$; $p=0.02$) and mRNFL ($F(2,215)=4.34$; $p=0.01$) (**Supplementary Table 7**). A Tukey-test revealed lower atrophy rates of pRNFL for SPMS compared to RRMS (diff= -0.39, $p=0.02$) and PPMS (diff= -0.43, $p=0.03$) and of mRNFL for SPMS compared to PPMS (diff= -0.26, $p=0.01$).”

To clarify how thickness change rates were calculated for the binned analysis to obtain the boxplots in **Figure 3** for subjects with more than two visits, we added the following sentence to the methods: “If more than two OCTs were in one interval, several annualized thickness change rates were entered for this subject in the interval.”

R1.3: In this context, fitting a mixed effects model to the population (with the dependent variable being the retinal layer thickness measurements and follow-up time as the independent variable, with a random intercept per eye nested in subjects and a random slope) and using the empirical Bayes estimates of best linear unbiased predictions (BLUPs) would be an approach with less sensitivity to outliers/random variability (see for example PMID: 21423039), and I suggest that this also be performed as a sensitivity analysis to assess the robustness of the findings. Notably, it seems that this may have been done, but this is described as including disease duration in the model and immediately following a discussion in the methods section outlining that thickness measurements were subtracted and divided by follow-up time.

A1.3: Thank you for this helpful comment. Following this comment, we have now computed empirical Bayes estimates of BLUPs of annualized thickness change rate and added the following sentences to the methods: “In order to account for the high variability of the thickness measurements, we performed empirical Bayes estimates of BLUPs of the longitudinal effect of disease duration (subtracting the baseline disease duration for each patient from the disease duration at assessment), reporting histograms of estimates across disease courses (**Supplementary Table 6**) and eye wise OLS with disease duration as only predictor and aggregated estimated regression coefficients (**Supplementary Table 7**). To decrease the high variability associated with the small number of observations in each stratum of disease duration, coefficients were estimated over the entire disease duration. To test differences between the disease courses, we ran an ANOVA on empirical Bayes estimates of BLUPs of annualized thickness change rates (**Supplementary Table 6**), using post hoc tests (Tukey) to determine which groups differed with respect to their atrophy rates.”

Moreover, we present the results as new **Supplementary Figure 4** and new **Supplementary Table 6** and have added the following section to the results: “Empirical Bayes estimates of best linear unbiased predictions (BLUPs) of annualized thickness change rates demonstrated significant atrophy of all retinal layers (pRNFL, mRNFL, GCIPL, and INL) for all groups (**Supplementary Figure 4**, BLUPs) and analysis of variance (ANOVA) significant differences across disease courses for pRNFL ($F(2,367)=3.65$; $p=0.03$) and mRNFL ($F(2,404)=4.43$; $p=0.01$) (**Supplementary Table 6**). A Tukey post-hoc test revealed higher atrophy rates of pRNFL for RRMS (diff=0.11, $p=0.02$) compared to SPMS and of mRNFL for PPMS (diff= 0.06, $p=0.01$) compared to SPMS.”

Reviewer #4:

R4.1: General: In many places, you report an estimate, confidence interval and p-value, but there are places with only the estimate and p-value. Please add confidence intervals in all places.

A4.1: We thank the reviewer for this important comment. We checked the entire manuscript and only identified missing confidence intervals in Supplementary Table 2. We calculated these missing values and added them to Supplementary Table 2.

R4.2: Page 6/Line 121 and Figure 2: In this analysis, it appears that you are fitting separate models for each of the groups and then stating which of the slopes are statistically significant. Could you compare the slopes between the groups using interaction terms because a difference in significance is not the same as a difference? As an example, the estimated slope in 2b was very similar for SPMS and PPMS groups, but one is significant, and the other is not. I think the conclusion here is that the association is similar rather than different between these groups.

A4.2: We thank the reviewer for this important comment. It is correct that we calculated a separate model for each of the groups. We have now clarified this in the methods section. We now include interaction terms and check their contrast-effects using f-tests.

We have added the following section to the methods: “An interaction term with disease duration and disease course as well as participants’ age, sex, and/or history of ON was included and its contrast-effects were checked using an F-test. If that test revealed significant differences, estimated marginal means of linear trends were used to test differences in atrophy rates between disease courses for significance (using Tukey’s method).”

We added the following section to the results: “Including an interaction term between disease duration and disease course, as well as participants’ age sex, and/or and history of ON and checking its contrast-effects using an F-test revealed that the effect of baseline disease duration on retinal layers, VA, and latency did not differ between groups (pRNFL: $F(3,464)=1.05$; $p=0.37$; mRNFL: $F(3,487)=0.99$; $p=0.40$; GCIPL: $F(3,491)=0.98$; $p=0.40$; INL: $F(3,423)=0.36$; $p=0.78$; VEP latency: $F(3,183)=0.26$; $p=0.86$; VA: $F(3,203)=1.55$; $p=0.20$).”

R4.3: Supplementary Table 2: As with the previous comment, could a model including an interaction term be used here to estimate the group specific values as well as compare the groups? I am also

uncertain why the regression coefficient for pRNFL in PPMS would decrease compared to Figure 2, but the p-value would get much smaller.

A4.3: We thank the reviewer for this helpful comment. We have now included interaction terms with disease duration and disease course as well as participants' age, sex, and/or history of ON and have checked their contrast-effects using f-tests (see also our response A4.2 to your previous comment).

We have added the following section to the methods: "An interaction term with disease duration and disease course as well as participants' age, sex, and/or history of ON was included and its contrast-effects were checked using an F-test. If that test revealed significant differences, estimated marginal means of linear trends were used to test differences in atrophy rates between disease courses for significance (using Tukey's method)."

Moreover, we have added the following section to the results: "Including an interaction term between disease duration and disease course, as well as participants' age sex, and/or and history of ON and checking its contrast-effects using an F-test revealed that the effect of baseline disease duration on retinal layers, VA, and latency did not differ between groups (pRNFL: $F(3,464)=1.05$; $p=0.37$; mRNFL: $F(3,487)=0.99$; $p=0.40$; GCIPL: $F(3,491)=0.98$; $p=0.40$; INL: $F(3,423)=0.36$; $p=0.78$; VEP latency: $F(3,183)=0.26$; $p=0.86$; VA: $F(3,203)=1.55$; $p=0.20$)."

The question concerning the observed decrease in the p value is no longer relevant because we had to correct the p and b values in Supplementary Table 2 after correcting for optic neuritis, participants' age and/or sex if the model fit could be improved.

R4.4: Page 7/Line 153: You state that significant loss of pRNFL occurred through the disease course in RRMS and PPMS, but the change was positive in the 5.6-7.5 year interval for RRMS. Is this sign correct in Supplementary Table 4?

A4.4: Yes, the change was positive in the 5.6-7.5 year interval for RRMS. We believe that this is the result of a statistical artifact rather than a true biological effect because the thickness changes rates for the other intervals and subgroups were generally all negative rather than positive (see Supplementary Table 4)

R4.5: Page 7/Line 157: As with my previous comments, you comment almost exclusively on the intervals with a p-value less than 0.05, but some of the intervals have only minor differences in the estimates despite large differences in the p-values. As an example, you state that mRNFL atrophy was observed in RRMS <3.5 years, but the estimated change is much smaller than in the 7.6-10.5 interval. It seems that a comparison between the groups/intervals or with the HCs would be more informative. It seems that models that included interaction terms would allow direct comparisons of the groups and intervals. Further, the comparison across the time intervals using a piecewise linear spline model would indicate whether the linear model described in Supplementary Table 3 is an appropriate fit.

A4.5: The samples sizes are very different between intervals and the courses of disease resulting in relatively unstable estimates which is the most likely reason for these results (see Supplementary

Table 5). Therefore, in line with reviewer 1, we performed the OLS to assess differences in effects of disease duration and compared slopes across different courses of disease using ANOVA.

Furthermore, we computed empirical Bayes estimates of BLUPs, reporting histograms of estimates across disease courses. In order to test differences between the disease courses, we ran ANOVAs on empirical Bayes estimates of BLUPs of thickness change rates, using post-hoc tests (Tukey) to determine which groups differ with respect to their atrophy rates. We have decided not to calculate piecewise linear spline models because they would provide the same information as the boxplots in Figure 3 and because the number of included eyes for each interval of disease duration by disease subtype was quite small (see Supplementary Table 5). We added the following section to the methods:

“In order to account for the high variability of the thickness measurements, we performed empirical Bayes estimates of BLUPs of the longitudinal effect of disease duration (subtracting the baseline disease duration for each patient from the disease duration at assessment) with time since baseline, age and sex as predictors, reporting histograms of estimates across disease courses (**Supplementary Table 6**) and eye wise OLS with disease duration as only predictor and aggregated estimated regression coefficients (**Supplementary Table 7**). To decrease the high variability associated with the small number of observations in each stratum of disease duration, coefficients were estimated over the entire disease duration. To test differences between the disease courses, we ran an ANOVA on empirical Bayes estimates of BLUPs of annualized thickness change rates (**Supplementary Table 6**), using post hoc tests (Tukey) to determine which groups differed with respect to their atrophy rates.

To further reduce the variability of OLS estimates, we discarded eyes with less than three OCTs (follow-up median (IQR): 3.95 years (2.97 – 6.03 years), **Supplementary Figure 5**). To test the aggregated coefficients for significance, Wilcoxon rank-sum tests were run for each outcome and disease course. To test whether there are significant differences between disease courses, we performed an ANOVA with random intercepts per subject. In cases where the ANOVA revealed significant group differences, we conducted a Tukey post-hoc test to identify which groups differed significantly.

While overall fixed effect estimates (**Supplementary Tables 3 and 4**) capture both cross-sectional and longitudinal effects, described BLUPs (**Supplementary Table 6**) and eye wise OLS (**Supplementary Table 7**) capture exclusively longitudinal effects of disease duration and therefore provide an additional perspective.”

R4.6: Page 9/Line 199: I am confused by the interpretation of the OR. You state that 1 microgram of atrophy is associated with an increase in the odds of disability progression by 38%. I think the OR would mean that a one unit increase leads to a 38% reduction in the odds but inverting the OR would be an OR of 1.62. Also, is this the OR for a relapse occurring or disability progression due to relapses?

A4.6: We thank the reviewer for this very important comment. In addition to the interpretation issues highlighted here, we realized that our input variables of the initial analysis were standardized, meaning that the odds ratios did not refer to micrometer differences, but to the sample standard deviations of the thickness measurements. We have now added the risk factor for the occurring of future relapses, MRI activity, and disability progression in the case of 1 μ m loss of pRNFL, mRNFL, GCIPL, and INL

thickness. Moreover, we have adjusted the interpretation in the discussion of the manuscript as: “As an example, 1 μm of pRNFL thickness loss in RRMS patients without history of ON increases the likelihood of relapse by 34% (Risk factor, Table 2, LMER).”

Furthermore, we have added the following sentence to the methods: “All continuous covariates were standardized in order to yield a more balanced optimization problem for model fitting.”

R4.7: Page 9: I do not understand why the results would be so different comparing the mixed effects logistic regression model and the Cox proportional hazards model for relapse with pRNFL (Table 2 vs. Supplementary Table 6). Could you explain the difference in the modeling assumptions since both approaches are analyzing the impact of pRNFL on the likelihood of a relapse?

A4.7: The cox regression only focusses on the first relapse in a time to event analysis while the logistic regression aims to predict whether a relapse will occur at a given time. Therefore, a patient who has a relapse very early and never again in the disease course, will get high hazard ratios in the cox framework, while the relapse likelihoods in the logistic regression are likely to be low.

R4.8: Page 9/Line 214 and Line 221: Were the inconclusive results not significant or were the confidence intervals too wide to be able to make a statement? I believe this should be clarified. If the results were just not significant, I think this is different than inconclusive.

A4.8: We thank the reviewer for this comment. We complemented the following sentence in the results: “The analyses did not reveal conclusive results (data not shown) as very wide confidence intervals resulted from a combination of the small sample size per stratum and the high variability in covariates.”

R4.9: Page 15/Line 405 and Table 1: Was a chi-squared test used to compare age between groups?

A4.9: We thank you for this important comment. We used a Kruskal-Wallis test to compare age between the different groups but incorrectly described this. We corrected the following sentences in the legend of Table 1 and methods: “^bKruskal-Wallis test; Pairwise comparisons revealed $p < 0.001$ for all comparisons except for PPMS vs SPMS ($p = 0.04$).” “Chi-square-test was conducted to compare sex and the number of included eyes with ON, with clinical history of ON, with history of ON identified by cut-off of GCIPL inter-eye difference ⁴³, and the number of patients on high-efficacy DMT and low-efficacy DMT between groups at baseline. Kruskal-Wallis-test was used to test for significant differences across disease subtypes at baseline regarding age, disease duration, and EDSS. A two-sided t-test was conducted to compare the follow-up time between groups.”

R4.10: Table 1: The listed IQR for the disease duration for RRMS patients seems to have an error since the IQR is from 0-30

A4.10: We thank you for this important comment. We have corrected all IQRs in Table 1. The previously reported IQRs were erroneously the ranges and not the IQRs (25th-75th percentile).

R4.11: Page 16: Reviewer 1 commented on the potential importance of distinguishing between cross-sectional and longitudinal effects, but the proposed model does not distinguish between these. Did

you test whether the longitudinal and cross-sectional effects of disease duration are equal as requested by Reviewer 1?

A4.11: We thank you for this comment. We had not explicitly tested whether the effects were based on cross-sectional or longitudinal effects but now provide additional analyses to present multiple perspectives. **Figure 2** and **Supplementary Figure 1** investigate only baseline and therefore cross-sectional differences. Our new empirical Bayes estimates of best linear unbiased predictions (BLUPs) and eye wise ordinary least squares (OLS) analyses (**Supplementary Figures 3 and 4**) capture exclusively longitudinal effects while the linear mixed-effects models (**Supplementary Tables 3 and 4**) pool both cross-sectional and longitudinal effects together. We have added the following section to the methods: "While overall fixed effect estimates (**Supplementary Tables 3 and 4**) capture both cross-sectional and longitudinal effects, described BLUPs (**Supplementary Table 6**) and eye wise OLS (**Supplementary Table 7**) capture exclusively longitudinal effects of disease duration and therefore provide an additional perspective."

Moreover, we have added the following section to the discussion to explain the differences and advantages of the different applied models to analyze the effect of time (disease duration) on retinal layer thickness: "In order to analyze the effect of time (disease duration) on retinal layer thickness, we present multiple approaches, each providing a unique perspective on estimation of retinal thickness change rates. With the baseline LMMs (**Figure 2**), we present an exclusively cross-sectional approach to estimate thickness change rates and complement it using non-linear spline regressions to capture potential non-linear thickness change rates (**Supplementary Figure 1**). Furthermore, we present two approaches that measure only longitudinal effects, with the BLUPs being based on an overall LMM (**Supplementary Figure S4, Supplementary Table 6**) and the eye-wise OLS on separate models for each eye (**Supplementary Figure S5, Supplementary Table 7**). Thus, BLUPs of thickness change rates consider a larger sample by also respecting overall thickness change rates, while the OLS approach focuses exclusively on estimating overall thickness change rates for each eye separately. Therefore, BLUPs provide a lower variance estimation of thickness change, which also causes smaller differences in estimates across patients. These differences in thickness change are better captured by eye-wise OLS, which in turn suffers from higher variance in estimates.

In addition to providing both complementary views on the estimation of thickness change rates (longitudinal and cross-sectional), we also present models that include both types of effects. Namely, we present a regular LMM (**Supplementary Table 3**) estimating linear thickness change rates across the entire sample as well as an LMM with the non-linear spline representation of the effect of disease duration (**Figure 3**). Therefore, our study provides an exhaustive analysis of retinal thickness change rates across MS patients of different subgroups."

R4.12: Page 16: Is your model including disease duration at assessment, which is changing with time, and age at assessment, which is also changing with time? This would seem to include two measures of within subject change. Also, it is also not clear which of the results in Supplementary Tables 3 and 4 included age and gender. Please clarify in the table legends which model was fit for each model.

A4.12: We thank you for this comment. Only disease duration was included as random effect in the model given the fact that both disease duration and age at assessment change at the same rate. For the analyses presented in **Supplementary Tables 3 and 4** the covariates were entered based on goodness of fit. We have now indicated where age and/or sex were included in the model and added the following sentence to the legend: “* age was included in the model; **sex was included in the model; *** age and sex were included in the model”. Each time results are presented, we now indicate which covariates were entered in the model.

R4.13: Page 16: I understand that the random effects were chosen based on likelihood ratio tests, but I am a little confused by the choice of the random effects. It seems that there is a random intercept for eye and no random slope for eye, and a random slope for subject but no random intercept for subject. Is that correct?

A4.13: No, that is not correct. A random intercept per eye was nested in subject, meaning, that there was both a random effect for eye and for subject. We thank the reviewer for this comment and realize that we should be more precise in the description of our statistical methods. We have added the following sentences to the methods: “In order to obtain parameter estimates for the annualized thickness change rate for the different intervals of disease duration (**Supplementary Table 4**) and for the overall annualized thickness change rate over time (**Supplementary Table 3**), retinal layer thickness was predicted based on the main effect of disease duration with a random intercept per eye nested in subjects and a random slope per subject using an LMM. A random slope was only introduced for subject, as, naturally, disease duration did not vary between eyes but only between subjects. Therefore, no random slope for eye was included in the model.”

R4.14: Page 16: When subjects had measurements that spanned multiple intervals, how was this handled?

A4.14: We thank you for this helpful comment. To clarify this point, we added the following sentence to the methods: “In cases with measurement intervals spanning several disease duration intervals the calculated annualized thickness change rate was considered for the interval of the follow-up scan.” This approach was used to not “water down” the more pronounced atrophy rates at the early phases of disease by averaging them with the slower atrophy rates at later phases. At the same time, we did not have any OCT intervals spanning from the very early to the very late phase of disease and therefore do not think that our approach bears the risk of substantially overestimating the atrophy rates in the later phases.

R4.15: Page 17, Line 452: Did you use a Gaussian link or an identity link?

A4.15: We hope that we understand this question correctly. In our model, we used the identity link function, and Gaussian errors. We did not use the Gaussian distribution function as a link function (probit model). We have added the following sentence to the methods: “These models were part of the gaussian family, calculated maximum likelihood based on Laplace approximation and modelled the

relationship between predictors and outcome variables with an identity link function, and Gaussian errors.”

R4.16: Page 17: For the analysis of EDSS progression, was the progression indicator (Y/N) a comparison of the EDSS to the previous EDSS measurement or the baseline EDSS measurement? Was the EDSS progression required to be sustained?

A4.16: We thank you for this important comment. We defined EDSS progression as difference of the actual EDSS to the previous EDSS. The EDSS increase did not have to be sustained. We changed the manuscript accordingly: “Disability worsening was defined as a documented increase in EDSS score compared to the previous measurement (≥ 1.0 point in case the EDSS score was < 6.0 , or ≥ 0.5 point if the EDSS score was ≥ 6.0) at a single time point. The EDSS increase did not have to be sustained.”

R4.17: Page 17: For the Cox model, was only time to first relapse analyzed or was it the time to each subsequent relapse? If there were multiple events, how were the recurrent events accounted for?

A4.17: We thank you for this comment. Only the time to first relapse was analyzed. Recurrent events were not entered in the time to event analysis. We added the following passage to the methods: “For the Cox model, only the time to first relapse, MRI activity or disability worsening (see methods) after baseline was analyzed. Recurrent events were not considered.”

REVIEWER COMMENTS

Reviewer #1 (Remarks to the Author):

I would like to thank and commend the authors for their efforts to address in detail the issues raised in my prior reviews. The manuscript has been strengthened significantly. I have no further comments.

Reviewer #4 (Remarks to the Author):

The paper has been improved with more information, but there remain a couple of aspects of the analysis that require more explanation.

Comments:

- 1) Page 6: It seems that you have added the interaction analysis, but you have left the original text as well. Therefore, you state that “mRNFL and GCIPL thickness were inversely associated with the disease duration in all subgroups except PPMS”, but this difference was not significant based on the interaction analysis. Is that correct?
- 2) Page 17: Why was a t-test used to compare follow-up time since there were more than two groups?
- 3) Page 19: It seems that a comparison of the slopes across the intervals would have been helpful for this section to see how the slope changed over time, but you state in the response to review that you chose not to do this because you included the box plots. The box plots do not provide a numerical comparison of the intervals.
- 4) Page 20: How is the Wilcoxon rank sum test used here? First, what groups are being compared here? Second, why is the nonparametric test needed compared to the ANOVA used for the comparing of the disease courses?
- 5) Page 21/Table 2: Please provide more information for the prognostic model mixed effects logistic model. Why is the sample size for the three outcomes so different in the “Prob of event” column and why is the sample size for the EDSS the smallest? Are you comparing the most recent OCT measurement to the presence or absence of each outcome in the next interval? Why are subjects without ON shown separately in this table? Please also clarify how the risk factor column was calculated.
- 6) Figure 5: Why do the curves not start at the same place? Kaplan-Meier failure curves usually start at 0, 0.
- 7) Supplementary Table 3: I believe that the results in this table need to be explained more carefully. It seems that you are estimating the median based on the within-subject analysis, but the b value is based on a mixed model with only disease duration which combines the cross-sectional and longitudinal effects. Is this correct? I think it is important to explain why the median for the mRNFL is equal to 0 for the RRMS and the SPMS (indicating no change with time), and the b value is significantly different from 0. The pRNFL estimates from this table also show a large difference between the median and the b values with the median not even being within the confidence interval for the b value. You explain the different models in the paper, but the table might lead to confusion.

Reviewer #4 (Remarks on code availability):

I reviewed three of the code files. I did not try to run the code myself, but I think the code would be usable for the community.

Reviewer #1:

R1.1: I would like to thank and commend the authors for their efforts to address in detail the issues raised in my prior reviews. The manuscript has been strengthened significantly. I have no further comments.

A1.1: Thank you for this positive feedback.

Reviewer #4:

R4.1: Page 6: It seems that you have added the interaction analysis, but you have left the original text as well. Therefore, you state that “mRNFL and GCIPL thickness were inversely associated with the disease duration in all subgroups except PPMS”, but this difference was not significant based on the interaction analysis. Is that correct?

A4.1: Yes, that is correct.

Figure 2 displays the linear regressions between retinal layer thickness, visual acuity and VEP latency at baseline and disease duration when adjusting for participants' age and/or sex if the model fit could be improved. Supplementary Table 2 shows the results of the linear regressions between retinal layer thickness, visual acuity, and VEP latency at baseline and disease duration when adjusting additionally for optic neuritis besides participants' age and/or sex if the model fit could be improved. An interaction term was only applied to the linear regressions when adjusting for optic neuritis, participants' age and/or sex as described in the methods: “An interaction term with disease duration and disease course as well as participants' age, sex, and/or history of ON was included and its contrast-effects were checked using an F-test. If that test revealed significant differences, estimated marginal means of linear trends were used to test differences in atrophy rates between disease courses for significance (using Tukey's method).” This is also described in the results: “Including an interaction term between disease duration and disease course, as well as participants' age sex, and/or and history of ON and checking its contrast-effects using an F-test revealed that the effect of baseline disease duration on retinal layers, VA, and latency did not differ between groups (pRNFL: $F(3,464)=1.05$; $p=0.37$; mRNFL: $F(3,487)=0.99$; $p=0.40$; GCIPL: $F(3,491)=0.98$; $p=0.40$; INL: $F(3,423)=0.36$; $p=0.78$; VEP latency: $F(3,183)=0.26$; $p=0.86$; VA: $F(3,203)=1.55$; $p=0.20$).”

R4.2: Page 17: Why was a t-test used to compare follow-up time since there were more than two groups?

A4.2: We thank you for this helpful comment. Previously, we used t-tests for pairwise comparisons of pairs of groups. In order to also provide information about general differences, we now used a Kruskal-Wallis-test and post-hoc Wilcoxon rank sum test to compare follow-up times at baseline between the different groups. We corrected this in the methods and in the legend of Table 1: “Kruskal-Wallis-test was used to test for significant differences across disease subtypes at baseline regarding age, disease duration, EDSS, and follow-up times.”⁹Kruskal-Wallis-test and post-hoc Wilcoxon rank sum test: HCs

vs PPMS ($p=0.49$), HCs vs RRMS ($p<0.001$), HCs vs SPMS ($p=0.27$), PPMS vs RRMS ($p<0.001$), PPMS vs SPMS ($p=0.27$), RRMS vs SPMS ($p=0.43$).”

R4.3: Page 19: It seems that a comparison of the slopes across the intervals would have been helpful for this section to see how the slope changed over time, but you state in the response to review that you chose not to do this because you included the box plots. The box plots do not provide a numerical comparison of the intervals.

A4.3: Thank you for this comment. Yes, the box plots are only shown for descriptive purposes. Due to the heterogeneity of the groups for the different periods of disease duration, with most patients displaying data points only in a few intervals and differences in the number of OCTs which were entered for a subject in an interval, adequate statistical comparisons between the boxes correcting for groups, intervals and within-subject factors did not seem constructive. Furthermore, we did not want to distract from the focus of the paper with complex analyses. Instead, we analyzed the effect of time (disease duration) on overall retinal layer thickness by using BLUPs, eye-wise OLS, and LMMs.

However, to address this reviewer’s comment we have included additional analyses (only) in this response letter: To test differences of retinal thickness change rates between the different disease duration intervals, we ran an analysis of variance (ANOVA) corrected for participants’ age and sex, using post hoc tests (Tukey) to determine which intervals differed with respect to their retinal thickness change rates. As a result of the above mentioned limitations we do not consider this analysis an adequate approach to investigate these data so we did not include it in the manuscript but only in this response letter:

The ANOVA showed significant differences of atrophy rates across the different periods of disease duration for pRNFL and GCIPL in RRMS and INL in PPMS (see the table below).

Analysis of variance on the annualized thickness change rates of the different layers for the different periods of the disease duration

RRMS	Df	F value	p value
pRNFL	5, 675	2.55	p = 0.03
mRNFL	5, 695	1.43	p = 0.21
GCIPL	5, 695	2.53	p = 0.03
INL	5, 695	1.40	p = 0.22
PPMS	Df	F value	p value

pRNFL	6, 258	0.74	p = 0.62
mRNFL	6, 272	1.22	p = 0.29
GCIPL	6, 272	0.81	p = 0.56
INL	6, 272	2.18	p = 0.045
SPMS	Df	F value	p value
pRNFL	5, 202	1.03	p = 0.40
mRNFL	5, 207	0.78	p = 0.56
GCIPL	5, 145	1.04	p = 0.39
INL	5, 207	0.44	p = 0.82

Legend: Df=number of degrees of freedom; pRNFL= peripapillary retinal nerve fiber layer; mRNFL=macular retinal nerve fiber layer; GCIPL=ganglion cell-inner plexiform layer; INL=inner nuclear layer; PPMS = primary progressive MS; RRMS = relapsing-remitting multiple sclerosis; SPMS = secondary progressive MS.

A Tukey post-hoc test revealed the following significant differences between the different boxes (all other comparisons non-significant): higher atrophy rates of pRNFL ($t=-3.38$, $p=0.01$) and GCIPL ($t=-3.31$, $p=0.01$) in the first interval of disease duration (0-3.5 years) compared to the third interval (5.6-7.5 years) in RRMS and higher atrophy rates of INL($t=-3.15$, $p=0.03$) in the third interval of disease duration (5.6-7.5 years) compared to the fourth interval (7.6-10.5 years) in PPMS.

R4.4: Page 20: How is the Wilcoxon rank sum test used here? First, what groups are being compared here? Second, why is the nonparametric test needed compared to the ANOVA used for the comparing of the disease courses?

A4.4: Thank you for this important comment. We have erroneously written that we used Wilcoxon rank-sum tests. However, we used one-sample t-tests to test if the ordinary least squares coefficients differed from zero. For details see line 69 in the code `SupFigure_5_SupTable7_Longitudinal_eyewise_OLS.R`.

We corrected the sentence accordingly in the methods: "To test the aggregated coefficients for significance, one-sample t-tests were run for each outcome and disease course."

R4.5: Page 21/Table 2: Please provide more information for the prognostic model mixed effects logistic model. Why is the sample size for the three outcomes so different in the “Prob of event” column and why is the sample size for the EDSS the smallest? Are you comparing the most recent OCT measurement to the presence or absence of each outcome in the next interval? Why are subjects without ON shown separately in this table? Please also clarify how the risk factor column was calculated.

A4.5: Thank you for this comment.

For clarification, we now provide more information about the prognostic mixed effects logistic model by adding the following section to the methods: “For the prediction of disease activity and disability progression (**Table 2**) all OCT measurements except the last OCT were included in the logistic mixed effects models. For the prediction of MRI progression/activity, the logistic regressions considered each of four possible MRIs between one OCT and the next OCT as a separate observation. Therefore, a single OCT could yield up to four observations (this was the maximum number of MRI observations, which one of our patients had). For the prediction of relapses, we aggregated all four possible relapses between one OCT and the next OCT. Thus, each OCT yielded a maximum of one observation for the analysis. Therefore, there are significantly fewer observations for relapses than for MRI. For the prediction of EDSS progression, the logistic regressions considered each EDSS (EDSS assessment was always performed with OCT measurement) after the considered OCT except the last one (since we cannot observe worsening after the last measurement). Thus, the analysis for EDSS considered one OCT less than for the analyses, resulting in a smaller sample size for EDSS.” Moreover, 29% of EDSS values were missing.

“We decided to show the results for patients without ON separately (**Table 2**)^{4,11-13} as retinal atrophy in the absence of ON may be considered a more suitable surrogate for chronic neurodegeneration and predictor for progression while the atrophy after ON mainly results from the presence and the severity of the inflammatory insult at the optic nerve”. We added this section to the discussion.

In our risk factor analysis, the odds ratio is the factor by which the risk of worsening is multiplied if the layer thickness is increased by one sample standard deviation (due to standardization of independent variables). In order to have a more interpretable estimate, we now also provide the risk factor. The risk factor means the factor by which the risk of EDSS progression, relapses, or MRI progression/activity increases for each micrometer of atrophy. Thus, risk factor = $\frac{1}{OR^{\frac{1}{\sigma}}}$

We also explained the calculation of the risk factor in the legend of Table 2: “Risk factor = factor by which the risk of EDSS progression, relapses, or MRI progression/activity increases for each μm thickness loss. Risk factor = $\frac{1}{OR^{\frac{1}{\sigma}}}$ “

R4.6: Figure 5: Why do the curves not start at the same place? Kaplan-Meier failure curves usually start at 0,0.

A4.6: We agree that it is unusual to have the Kaplan-Meier curves not start at 0.0. This is not a mistake, since there were simply no patients that were censored at time 0. Still, we agree that this might cause confusion and uploaded a new version of Figure 5 with lines extended to 0.0.

R4.7: Supplementary Table 3: I believe that the results in this table need to be explained more carefully. It seems that you are estimating the median based on the within-subject analysis, but the b value is based on a mixed model with only disease duration which combines the cross-sectional and longitudinal effects. Is this correct? I think it is important to explain why the median for the mRNFL is equal to 0 for the RRMS and the SPMS (indicating no change with time), and the b value is significantly different from 0. The pRNFL estimates from this table also show a large difference between the median and the b values with the median not even being within the confidence interval for the b value. You explain the different models in the paper, but the table might lead to confusion.

A4.7: Thank you for this important comment. The former Supplementary Table 3 demonstrates the results of two analyses which are entirely different and serve different purposes. In order to avoid misunderstanding, we have changed the former Supplementary Table 3 and now present the median and IQR of overall annualized thickness change rates over time in a different supplementary table (**Supplementary Table 3**) than the p and b values (**Supplementary Table 4**, LMM). Supplementary Table 3 now descriptively shows overall median and IQR while the Supplementary Table 4 now shows coefficients of the LMM corrected for participants age and/or sex and including random effects to adjust for nested data structure. We changed the numbering of the supplementary tables accordingly in the manuscript and the supplementary material.

REVIEWERS' COMMENTS

Reviewer #4 (Remarks to the Author):

The authors have addressed my concerns. I would like to thank the authors for being very responsive to my comments.